# Poverty induced inequality in nutrition among children born during 2010–2021 in India

**Junaid Khan**[1]*, **Sanjay K. Mohanty**[2]

**1** Department of Statistics, Vivekananda College, Thakurpukur, Kolkata, India, **2** Department of Population & Development, International Institute for Population Sciences, Mumbai, Maharashtra, Indiaa

* statjun@gmail.com

**Data Availability Statement:** The minimal dataset is made available through https://doi.org/10.7910/DVN/OUXYIX.

**Funding:** The author received no specific funding for this work.

## Abstract

### Introduction

Almost two-fifth of the children in India is stunted and among various factors, poverty differential in child undernutrition is the largest. Using the latest population-based survey of National Family Health Survey, 2015–16 and 2019–21 this paper examined the poverty induced inequality in child stunting across the sub-populations of India.

### Methods

A sample of 213,136 children aged between 0–5 years from NFHS fourth round and 98,222 children in the same age group from the NFHS fifth round constitute the study sample. The wealth index is used as the proxy of household's economic wellbeing and height-for-age (HAZ) z-score of a child is used to identify the stunting status of the child. Box plots are drawn to understand the distributional characteristics of the HAZ score for both the study sample. We calculate the Erreygers corrected concentration index and decomposed the concentration indices using Gonzalo-Almorox and Urbanos-Garrido method.

### Results

During 2015–16, more than half of the children from the poorest wealth quintile were stunted (52%), compared to 22% among the children from richest wealth quintile. In 2015–16, stunting was as high as 65% among the children of mothers with low stature (height less than 145 cm) and from the poorest wealth quintile whereas, the prevalence was observed 56% from the same sub-population during 2019–21. Among various factors, the concentration index of stunting was observed highest among the children of 36–47 months (-0.28) followed by children of age 48–59 months (-0.27) and among the fully immunized children (-0.25). Similar to NFHS-4, NFHS-5 also shows a predominantly higher socio-economic inequality among 24+ months children and among the fully immunised children. Factors like child age, birth order and sanitation showed positive elasticity. Decomposition analysis of NFHS-4 data shows that due to uneven distribution of wealth, mother's education as a determinant of child stunting solely explained 33% of the overall inequality followed by improved access to sanitation (24%), mother's height (8%) and place of residence (5%). Similar to NFHS-4, NFHS-5 data also shows that mother's education, sanitation, mother's

**Competing interests:** The authors have declared that no competing interests exist.

**Abbreviations:** NFHS, National Family Health Survey; CI, Concentration index; HAZ, Height-for-Age z score; LFA, Length-for- Age; SD, Standard Deviation; HDI, Human Development Index; DHS, Demographic Health Survey; MOHFW, Ministry of Health and Family Welfare; PSU, Primary Sampling Unit; CEB, Census Enumeration Block; PCA, Principal Component Analysis; RO, Reverse Osmosis; GLM, Generalised Linear Modelling; SC, Scheduled Caste; ST, Scheduled Tribe; ECI, Erreygers Concentration Index; ICDS, Integrated Child Development Service; UNDP, United Nations Development Programme; NRHM, National Rural Health Mission; MRD, Ministry of Rural Development.

height and place of residence predominantly contributes to the overall wealth inequality in child stunting.

## Conclusions

In India, poverty differential in child undernutrition is acute among the different sub-population of children. And the concentration of stunted children is higher among the different sub-population with higher wealth poverty. Mother's education, improved sanitation and mother's height explained larger variation in the overall inequalities in child nutrition across India.

## Introduction

Though there are different forms of child malnutrition, undernutrition is much more prevalent in developing countries, stunting is the most severe form of undernutrition among the children and has the severe effect on child's survival, health and overall physical development [1, 2]. Due to household poverty, poor diet practice, poor environment and poor care during pregnancy, the foetal/child growth remains poor [1]. Stunting is the form of chronic undernutrition which is defined as the height-for-age z-score less than minus two standard deviation (-2SD) below the median of a reference standard population [3]. The measure of stunting identifies the shortness in achievable height among the children under age five for his/her age and informs about the nutritional history [4].

South Asia is home to 58.7 million stunted children and carries almost half of the (47%) of the global burden of stunting among the preschool children [5]. The Global Nutrition Targets of the 2030 Sustainable Development Agenda aim to reduce the burden of undernutrition by 40% by the end of 2025 [6]. India being the second most populous country, constitutes the largest share of the child population and stunted children worldwide [7, 8]. Over the last two decades, India has shown a decrease (52% during 1992–93 to 38% during 2015–16) in the prevalence of child stunting but the variation is high across the states and districts of India [8, 9]. With a national average of 38%, the prevalence of stunting children is lowest (20%) in Kerala and highest (48%) in Bihar whereas the sub population of India conceals a large variation. Previous studies show detrimental evidences on child stunting and its association with the socio-economic characteristics [10–12]. And according to the Global Nutrition Report, globally 149 million children are stunted [13].

India still stands as the development paradox and in terms of overall human development index (HDI) ranks 130 globally though it has made an improvement in terms of absolute reduction in poverty, increase in life expectancy and improvement in education and standard of living but failed in terms of social progress in the last decade [14–17]. However, the inequality in developmental parameters like health, education and health care access are large within the country [18, 19]. At the same time, it is also evident that it is the poor with low income show poor health status and malnourishment among those children from the poor families [20, 21].

Previous studies showed that nutritional status among the children under age five in the developing countries vary largely across the sub population and household's socioeconomic characteristics, sanitation condition, place of residence, mother's education and height largely determines the nutritional status [22–27]. Thus, studies exploring the inequalities in population health are important to find the health gap across the heterogeneous population within a particular country or across countries. And the inequality in health could be due to economic

wellbeing or due to social gap [25, 28–30]. Most of the previous studies based in developing country settings focused on poverty induced inequality in population health and a large number of studies examined the distribution of income and its association with the health parameters and suggested higher income difference leads to lower standards of population health [31, 32]. A Systematic review of studies reported increase in the number of studies measuring health inequalities in different domains of population health like—mortality, communicable and non-communicable diseases, nutrition, mental health, risk factors and injuries where wealth status and income is commonly used as the measure of equity followed by educational attainment and gender bias [33]. A number of studies examined the related socio-economic inequality specific to child health parameters and used regression-based decomposition to measure the wealth poverty induced inequalities in child health [34–36]. In this context, this study adds to the existing knowledge in the domain of wealth-based inequality in child stunting using the most recent round of National Family Health Surveys for India. The study used the Erreygers corrected measure of concentration indices to measure the concentration of stunting prevalence across the population sub groups subject to wealth poverty and decomposed the concentration indices by socio-demographic characteristics using Gonzalo-Almorox and Urbanos-Garrido method.

## Methods

### Data

We utilised the data of children aged 0–59 months from the National Family Health Survey (NFHS)-2015-16 and 2019–21. NFHS is one of the important surveys in India which provides necessary information on child health, most importantly child nutrition-stunting, underweight and wasting along with other health and household indicators and the survey estimates are most commonly used among the policy makers and among the government and non-governmental institutions to introduce new health intervention in the population. NFHS, 2015–16 is the fourth round in the series of DHS in India and all the NFHS rounds contribute to the demographic database for India. As the study is based on secondary data available in public domain for research; no ethical approval was required from any institutional review board (IRB).

### Sample size and sampling

NFHS, 2015–16 survey adopted a multistage sampling strategy and used the 2011 Census India sampling frame. The villages in the rural areas and census enumeration blocks in the urban areas served as the primary sampling units (PSUs) for the first stage of sample selection. A total of 28,586 PSUs were selected randomly across the country and the fieldwork was completed in 28,522 clusters. Households within the PSUs served as the second stage sampling units. PSUs with fewer than 40 households were merged to the nearest PSU. Using probability proportional to size sampling, villages were selected from each rural stratum and in the urban areas, census enumeration blocks (CEBs) were selected from the list of CEBs, obtained from the Office of the Registrar General and Census Commissioner; New Delhi. Prior to the main survey, a complete household mapping and listing was done in the selected rural and urban PSUs. And within the selected PSUs (consisting of 300 or more households), households were divided into segments of 100–150 households. And finally, two of the segments were selected using systematic sampling with probability proportional to segment size.

NFHS-5 also adopted a multistage sampling strategy and used the 2011 Census India sampling frame. NFHS-5 collected the information from 707 districts, 28 states, and 8 union territories. During the NFHS-5 survey, each district was divided into urban and rural strata. The

rural stratum was further subdivided into smaller substrata based on village population and the proportion of the population from Scheduled Castes (SC) and Scheduled Tribes (ST). In each rural stratum, villages were chosen as Primary Sampling Units (PSUs), with PSUs sorted by the literacy rate of women aged 6 and above prior to selection. For the urban strata, Census Enumeration Blocks (CEBs) were chosen as PSUs, with sorting based on the percentage of SC/ST population. In the second stage, a fixed number of 22 households per cluster were systematically selected with equal probability from a newly created list of households within the chosen PSUs. This household list was generated through mapping and listing operations conducted in each selected PSU before the second stage of selection. In total, 30,456 PSUs were selected from 707 districts nationwide for NFHS-5.

### Variables and measurements

**Outcome variable.** The main outcome variable of the study is stunting which is an anthropometric measure of child's nutritional status. To measure the nutritional status among the children, children's height (length for children under age 2 years), weight and age in months were used and the anthropometric measure of child's nutritional status i.e., length-for-age (LFA) Z-score for children under age 2 years and height-for- age (HFA) Z-score for those children above age 2 years were generated. To define stunting among the study children, the WHO, 2006 child growth standard was referred for both LFA as well as for HFA Z scores. As both the scores LFA and HFA help to identify the impaired linear growth among the children under age two and above two respectively, we used the term stunting commonly throughout the study. During the survey of NFHS-4, Infantometer (Seca 417) was used to measure the recumbent length of the children under age two years and Stadiometer (Seca 213) was used to measure the height of the children aged 24–59 months. Seca 874 scale was used to measure the weight of the children.

*Wealth poverty*. NFHS collected the asset based (consumer goods like owning of television, bicycle, radio etc.) information from the randomly selected households and provided the wealth index as a proxy measure on household's economic wellbeing [37]. A set of 37 asset-based indictors were used to create the wealth score for the households using a principal component analysis (PCA). And any member of the same household was assigned the same status of economic wellbeing, the household belongs to. The five economic classes of wealth quintiles are- poorest, poorer, middle, richer & richest.

**Correlates.** Child's individual characteristics, maternal characteristics and the household's characteristics are considered as the potential correlates for child stunting. Table 1 depicts the variables included in this study along with their categories. Child's characteristics include socio demographic and economic characteristics, while community- level variables include the common characteristics of study subjects in an enumeration area such as region and place of residence. The description of the correlates is as follows- **Age of the child:** age of the child is captured in months and categorised as 0–5, 6–11, 12–23, 24–35, 36–47 & 48–59 months (for NFHS-5, the last three categories are merged and defined as 24+ months). **Gender of the child:** gender of the child (male/female) is a dichotomous variable and a bio-demographic identification of the child. **Birth order of the child:** birth order of a particular child is categorised into four categories as- 1, 2–3, 4–5 & 6+. **Immunization status of the child:** child's immunization status is a key variable which signifies the utilization of health care services for vaccination by parents for their children. Child's immunization status has been compiled from the data which has been categorised as–no, partial and full immunization. **Mother's educational attainment:** mother's educational attainment has been categorised into four categories and are as follows- no education, primary completed, secondary completed and higher

**Table 1. Percent distribution of children under five years by the population characteristics, India, NFHS, 2015–16 & 2019–21.**

| Variables | NFHS-4 | NFHS-5 |
|---|---|---|
| | Distribution (%) | Distribution (%) |
| **Age in months** | | |
| 0–5 | 7.95 | 15.7 |
| 6–11 | 10.02 | 16.3 |
| 12–23 | 20.05 | 33.3 |
| 24–35 | 20.13 | 34.78* |
| 36–47 | 21.33 | NA |
| 48–59 | 20.52 | NA |
| **Sex of the child** | | |
| Male | 51.76 | 51.4 |
| Female | 48.24 | 48.6 |
| **Birth order** | | |
| 1 | 36.05 | 37.6 |
| 2–3 | 47.41 | 49.0 |
| 4–5 | 12.36 | 10.6 |
| 6+ | 4.18 | 2.8 |
| **Immunization** | | |
| No | 9.56 | 4.8 |
| Partial | 39.31 | 48.1 |
| Full | 51.13 | 47.1 |
| **Mother's Education** | | |
| No education | 31.17 | 20.5 |
| Primary completed | 14.57 | 12.3 |
| Secondary Completed | 45.24 | 52.7 |
| Higher | 9.01 | 14.5 |
| **Mother's height** | | |
| <145cm | 11.7 | 11.9 |
| > = 145 cm | 88.3 | 88.1 |
| **Residence** | | |
| Rural | 76.25 | 80.2 |
| Urban | 23.75 | 19.8 |
| **Sanitation** | | |
| Improved | 50.18 | 74.2 |
| Unimproved | 49.82 | 25.8 |
| **Drinking water** | | |
| Improved | 87.53 | 87.9 |
| Unimproved | 12.47 | 12.1 |
| **Wealth Quintile** | | |
| Poorest | 26.12 | 26.9 |
| Poorer | 23.63 | 23.0 |
| Middle | 20.05 | 18.9 |
| Richer | 16.71 | 17.0 |
| Richest | 13.49 | 14.2 |
| **Regions** | | |
| North | 18.92 | 18.9 |
| Central | 28.73 | 26.1 |
| East | 21.11 | 20.4 |

(*Continued*)

**Table 1.** (Continued)

| Variables | NFHS-4 | NFHS-5 |
|---|---|---|
| | Distribution (%) | Distribution (%) |
| North East | 14.86 | 16.0 |
| West | 6.77 | 9.2 |
| South | 9.61 | 9.3 |
| **N** | **2,13,136** | **98,222** |

*24–59 months old children population

educated. **Mother's stature:** Maternal stature has been directly utilised as an indicator of maternal anthropometry and has been categorised in a dichotomous way. Mother's with less than 145 cm of stature are considered as stunted and otherwise. **Residence:** Place of residence of the children is another binary demographic variable and the categories are rural/urban. **Toilet facility:** As per the definition of NFHS-4, access to toilet facility has been categorised as improved and unimproved toilet facility. **Drinking water facility:** Similar to toilet facility, access to drinking water has also been categorised following the definition of NFHS-4 and households with access to any of these sources-piped water, public tap/standpipe, tube well or borehole, protected dug well, protected spring, rainwater, community RO (Reverse Osmosis) plant are classified as improved source of drinking water and unimproved otherwise. **Regions:** As per the NFHS definition, Indian states (the first administrative unit) are grouped into six different regions namely, North, Central, East, North-East, South and West.

**Statistical analyses.** A total of 213,136 children aged 0–59 months were analysed in this study. Taking care of the survey design, all the estimates provided are weighted.

Of the total 259,627 cases there were 23,172 cases for which the anthropometric information was not available. For the rest of the children, there are 10,235 cases found to be flagged, 21 cases for which the age in days is out of the plausible limits and there are 1,197 cases for which the height information is found to be out of the plausible limits. So, the analytical sample of this study is 225,002. Considering the completeness of the information on the outcome variable and the independent predictors, we found a total of 213,136 children constituting the analytical sample of the study. Similar to NFHS-4, and considering the availability of the pertinent information through NFHS-5, a total of 98,222 children constitutes the study sample.

The standard measure of socio-economic inequality for any health indicator for a particular population [38, 39] is given by the following-

$$Concentration\ Index\ (CI) = \frac{2}{\mu} cov(\gamma_i, R_i) \tag{1}$$

Where μ is the mean of stunted children, $\gamma_i$ is the nutritional status of the i-th children and $R_i$ is the cumulative percentage that the i-th child represents over the total population once the i-th child ranked by socio-economic status here in this study by wealth score. The value of concentration index ranges between -1 to +1. A negative value implies that stunting is more concentrated among the poor population while a positive value indicates that stunting is more concentrated among the rich population. And the value zero denotes the perfect example of no wealth related inequality in the health measure of the children.

Here in this study, we do not use the conventional measure of concentration index to measure the socio-economic inequality rather we introduce the measure Erreygers corrected concentration index taking care of the binary nature of the child's nutritional status variable [40]. Because, the conventional measure of concentration index does not take care of the bounded

nature of the health variable. When the outcome variable is dichotomous in nature, Erreygers suggests a correction in the concentration index multiplying the CI value with a factor to allow the comparison between two different sub-populations with different average heath (nutritional) status. The expression of the Erreygers corrected concentration is given as following-

$$E(y) = \frac{4\mu}{y^{max} - y^{min}} CI \tag{2}$$

Where $\mu$ is the mean prevalence of stunting and $y^{max}$ and $y^{min}$ are the two extreme values. To decompose the concentration indices of a binary variable subject to wealth poverty we utilised the generalised linear modelling (GLM) of a binomial family with probit link function which is recommended and provides the consistent estimate of the effects of demographic and socio-economic factors on child stunting independent of the choice of reference category [41, 42]. When there is a linear relationship between the outcome and the set of explanatory variables, the concentration index can be expressed as the weighted sum of the partial concentration indices for the explanatory factors of inequality and thereby the changes in CI due to a particular factor can be disentangled by Oaxaca-type decomposition method [43]. While dealing with a dichotomous variable, non-linear models are apt with a maximum likelihood approach and the decomposition is done with the help of a linear approximation [44]. As we utilised the Erreygers corrected concentration index to measure the inequality, we employed the Gonzalo-Almorox and Urbanos-Garrido method [45] of decomposing and the expression is given as below-

$$E(y) = 4. \sum_j (\beta_j^k \bar{x}_j) CI_j + GCI_\varepsilon \tag{3}$$

Where E(y) is the Erreygers corrected concentration index, $\bar{x}_j$ is the mean of the j-th explanatory variable, $CI_j$ is the average value of the j-th variable, $\beta_j^k$ is the partial effect for the k-th category of the j-th variable and $GCI_\varepsilon$ is the generalised concentration index of the error term. The estimated partial effects from the probit model are used to compute the contributions of the explanatory variables considered in the study framework. In summary, the factor level contributions are calculated as follows: first, the partial effects calculated for each $x$. Second, the mean of the outcome variable and the elasticity of the outcome variable are calculated with respect to each $x$. Third, the CIs are calculated in terms of each $x$. Fourth, the contribution of each $x$ in CIs is calculated by multiplying the elasticity of the variable by its CI. The decomposition of CI provides the estimates on elasticity, absolute contribution and percentage contribution in the overall inequality. Statistical analysis is performed using STATA version 14.1 (StataCorp™, Texas).

## Results

### Descriptive statistics

The dataset was complete in terms of all the concerned variables included in the analyses. Table 1 showed description of the children included in the study. Among the study children, 20% belonged to the 12–23 months of age group with lowest (8%) in the 0–5 months of age group. The mean age of the children was 30 months. Fifty-two percent of the total children were male. Almost two-fifth (37%) of the study children were of first birth order. Only, half of the children were fully immunized and one-tenth of the children did not receive any of the doses of full immunization. Mothers to 31% of the children had no formal education. Twelve percent of the children's mothers were found to be less than 145cm in height. Seventy six percent of the children were rural and half of the children did not have access to improved

sanitation facility. Twelve percent of the children did not have the access to safe drinking water. Around 41 percent of the study children belong to one of the socially excluded groups-scheduled caste or scheduled tribe [46, 47]. Half of the children belong to the lowest two wealth quintiles (Poorest and Poorer).

The NFHS-5 data shows that the majority of children (33.3%) are aged between 12–23 months, with a nearly equal gender distribution (51.4% male, 48.6% female). Most children are either first-born (37.6%) or second/third-born (49%). Regarding immunization, 47.1% of children are fully immunized, while 48.1% have partial immunization, and only 4.8% remain unimmunized. Mothers of these children are predominantly educated up to secondary level (52.7%), and most have a height of 145 cm or above (88.1%). The majority of families reside in rural areas (80.2%) and have access to improved sanitation (74.2%) and drinking water (87.9%). In terms of wealth, 26.9% belong to the poorest quintile, while only 14.2% are in the richest category, with the Central region having the highest share of the population (26.1%).

## Distribution of HAZ scores

Box plots were drawn for the each of the explanatory variables to understand the distributional characteristics of the HAZ score as well as skewness and data quartiles of the scores given the categories of selected variables. The line that divided the box into two parts showed the mid-point of the data which was the median value of the z-score. Apparently, the z-scores of half of the children population fell below the median z-score. Figs 1–11 showed the HAZ scores for all the concerned variables. Fig 1 showed the distribution of the z-score by age groups which demonstrated the shape of the boxes is almost uniform and short indicating a less variation in the score within every age group of children. Fig 2 depicted the pattern of HAZ score by birth order of the children and the boxes were observed to be not very short which suggested that children within a particular group of the defined sub population (by birth order) had quite

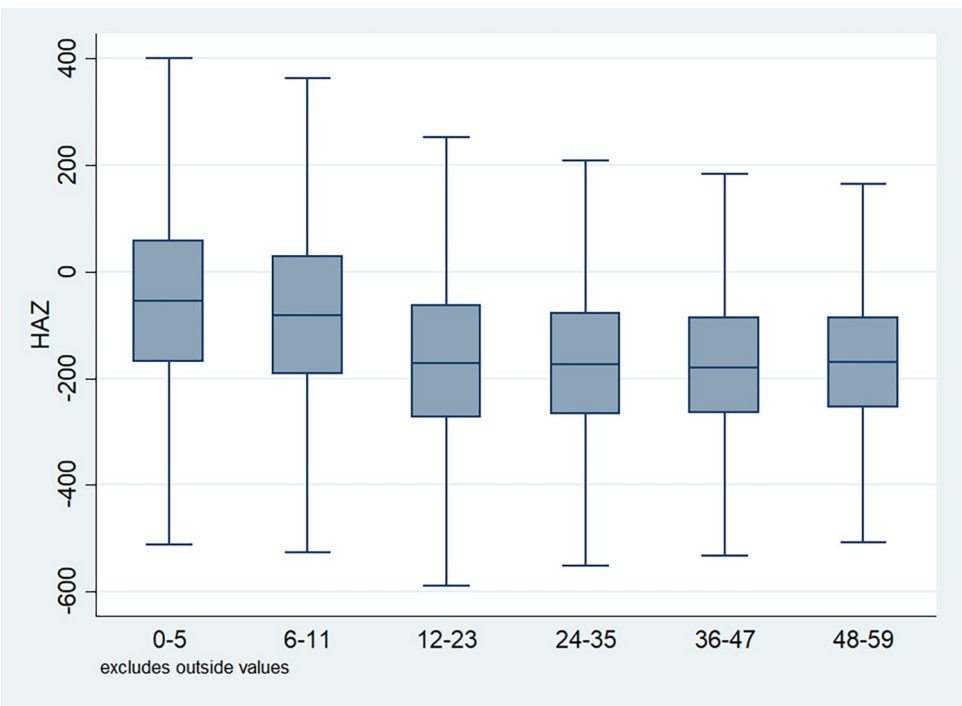

**Fig 1. Box plot of HAZ score by age of the child, NFHS, 2015–16, India.**

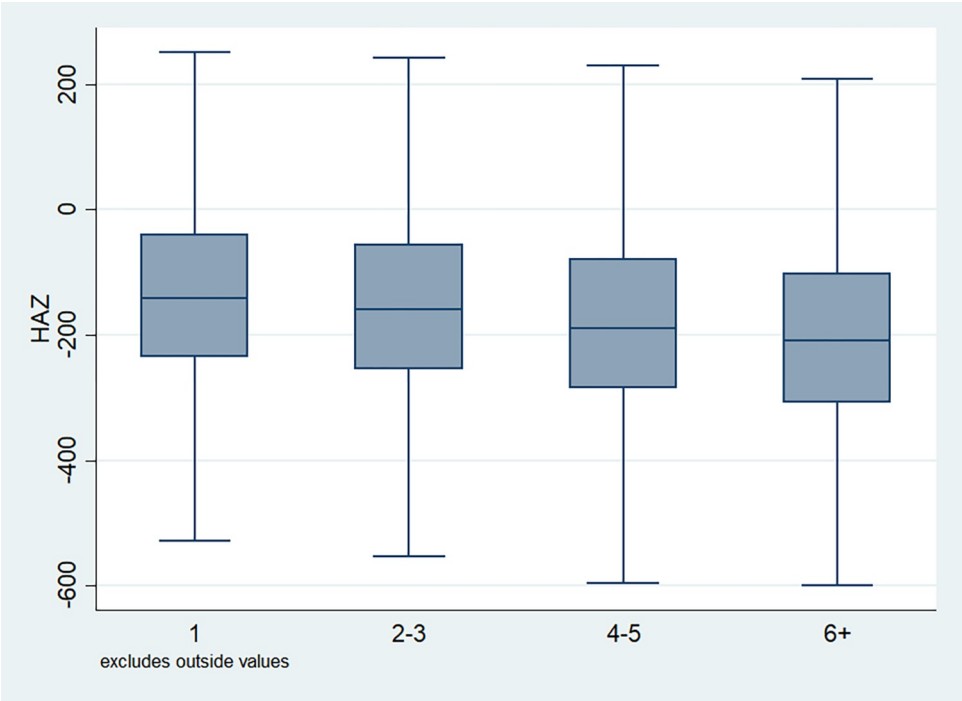

**Fig 2. Box plot of HAZ score by birth order of the child, NFHS, 2015–16, India.**

different z-scores among them. This showed the differential in HAZ score among the children of a specific birth order.

It is also observed that with increasing birth order, the HAZ score for higher number of children fell below the score -200. The shape of the box for full immunization was observed shorter than those children with no immunization (Fig 3). By mother's education, the HAZ scores of the children were observed to vary in different ranges and higher the educational attainment of the mothers, the median value was also found to be increasing (Fig 4). Fig 5 showed that lower the stature of mothers, children of those mothers showed lower HAZ scores among them. Similarly, Fig 6 described the regional distribution of HAZ scores among the children. Children from the Central and Eastern part of India showed low HAZ scores with higher percentage of children scoring below the -200 HAZ score. Similarly rest of the figures explained the nature of HAZ score among the children by residence, gender, toilet facility, access to improved drinking water and wealth quintile. Similar to NFHS-4, the NFHS-5 patterns were also shown (S1 Appendix)

## Prevalence of stunting by wealth quintile and background characteristics

Table 2 presented the estimated prevalence of stunting by wealth quintile and by background characteristics. These estimates provided the pattern of stunting prevalence among the heterogeneous child population subject to wealth poverty. Within each of the wealth stratum, there remains large variation in the stunting prevalence by child's population characteristics- age of children, birth order, mother's education, mother's height and across regions. At the national level, wealth status does show a clear distinct differential in the stunting prevalence and children from the poorest wealth quintile showed the highest prevalence (51%) with a gradual decrease over the higher wealth quintiles. Children of age 24–39 months carried the highest

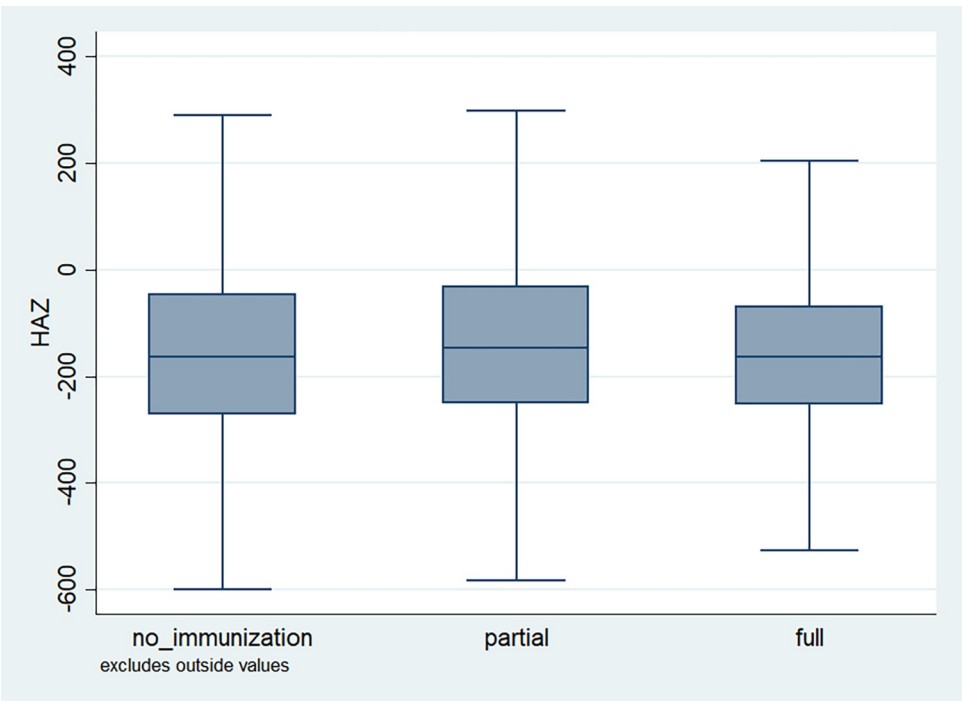

**Fig 3. Box plot of HAZ score by immunization status of the child, NFHS, 2015–16, India.**

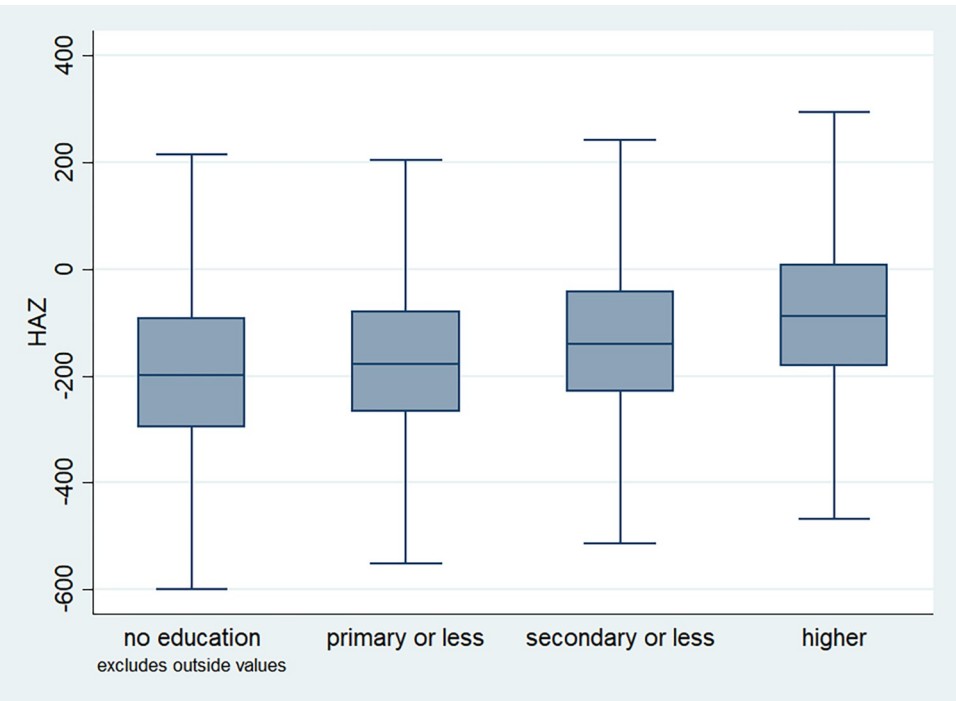

**Fig 4. Box plot of HAZ score by mother's education of the child, NFHS, 2015–16, India.**

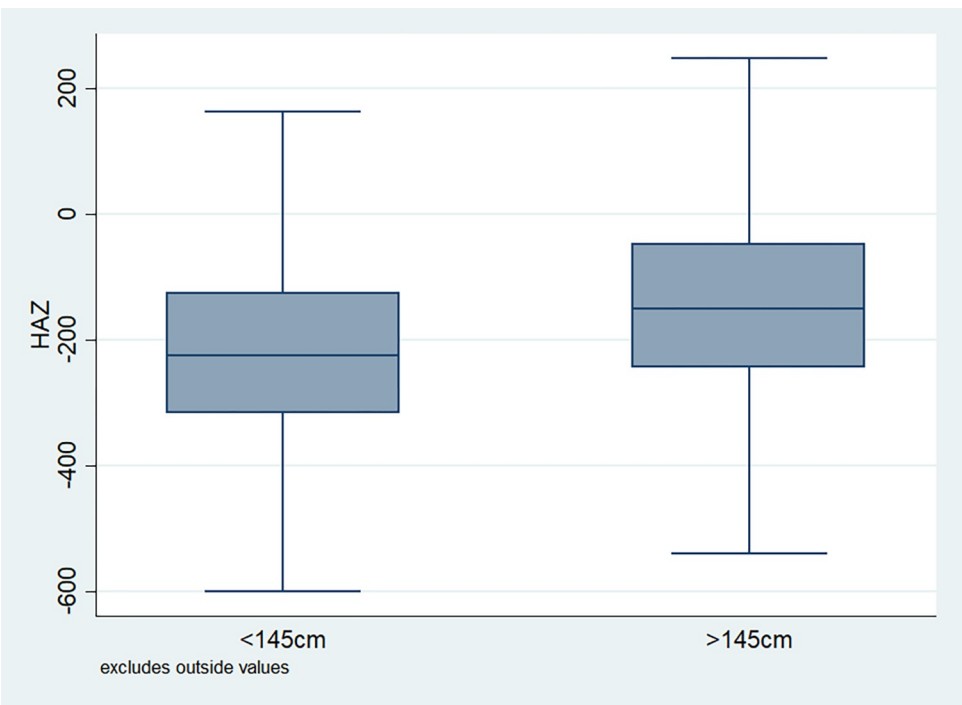

**Fig 5. Box plot of HAZ score by mother's height of the child, NFHS, 2015–16, India.**

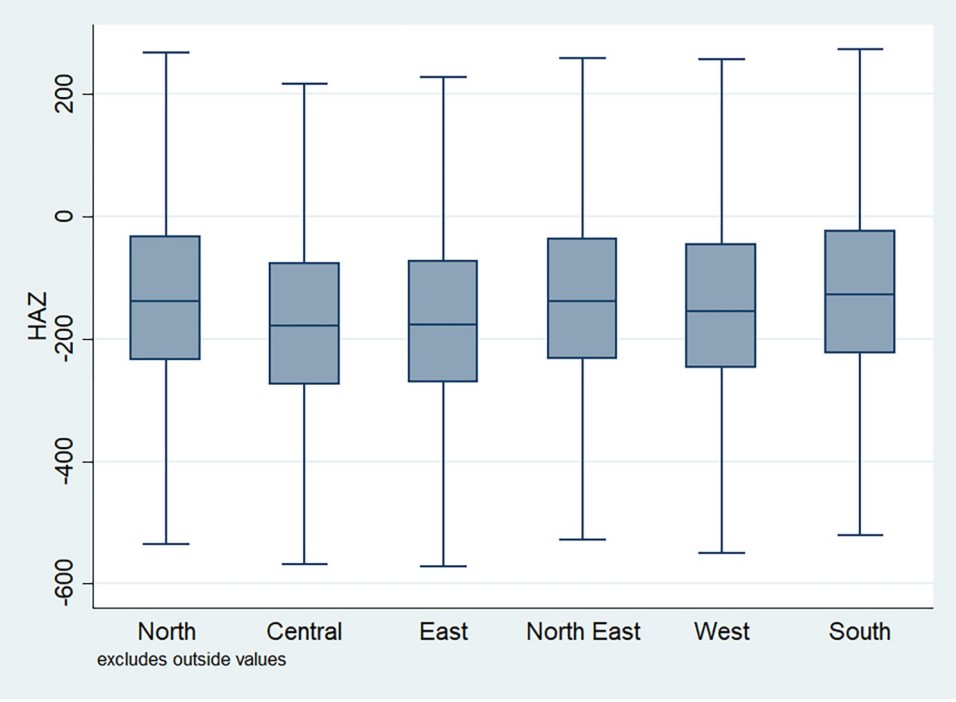

**Fig 6. Box plot of HAZ score across regions, NFHS, 2015–16, India.**

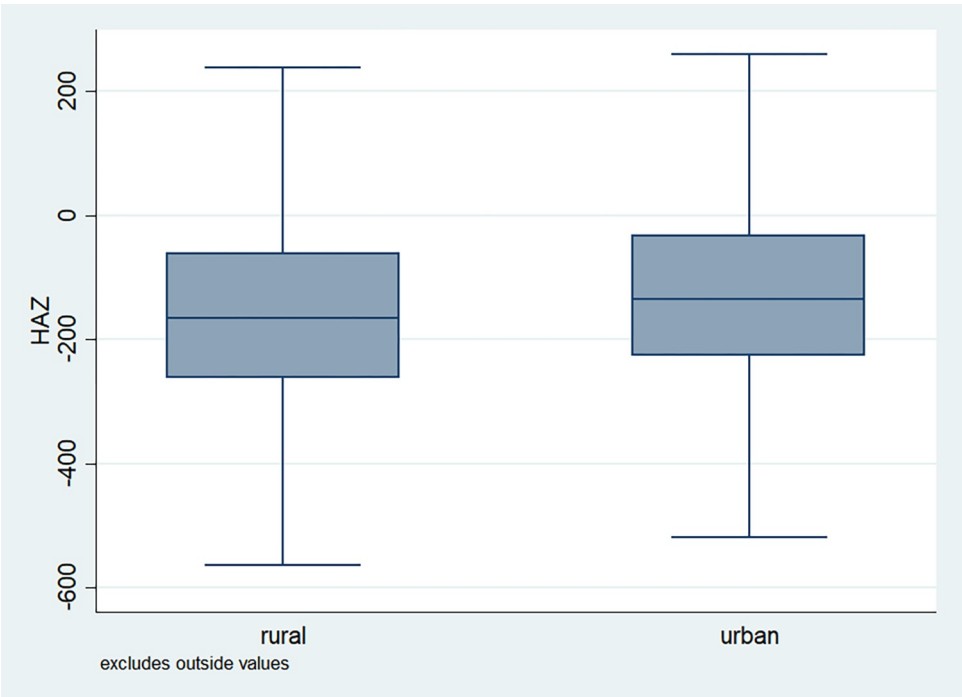

**Fig 7. Box plot of HAZ score by place of residence, NFHS, 2015–16, India.**

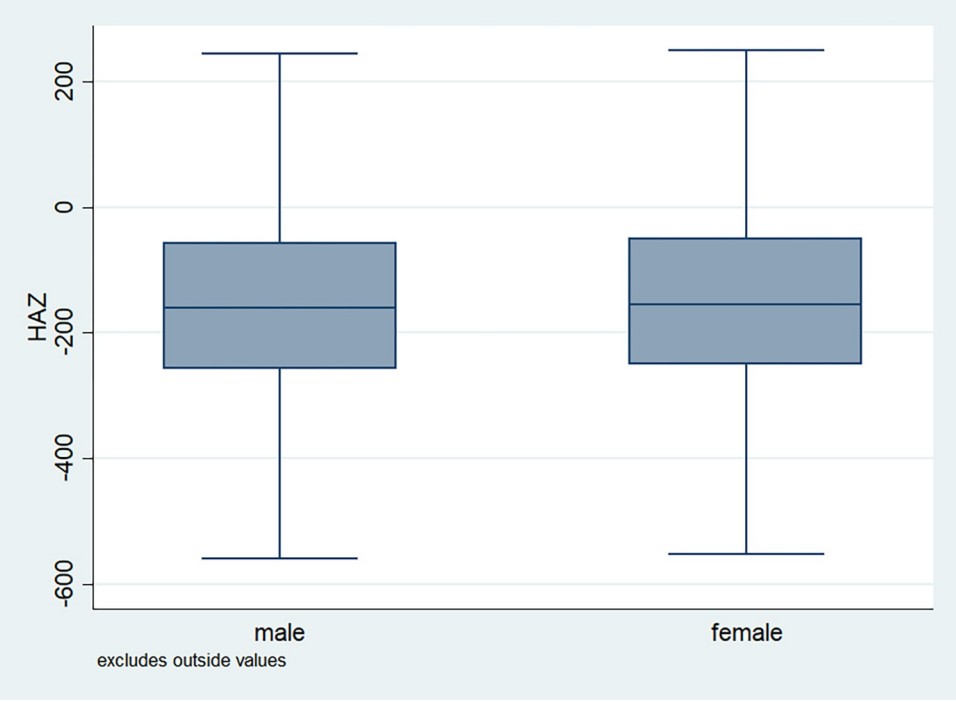

**Fig 8. Box plot of HAZ score by gender of the child, NFHS, 2015–16, India.**

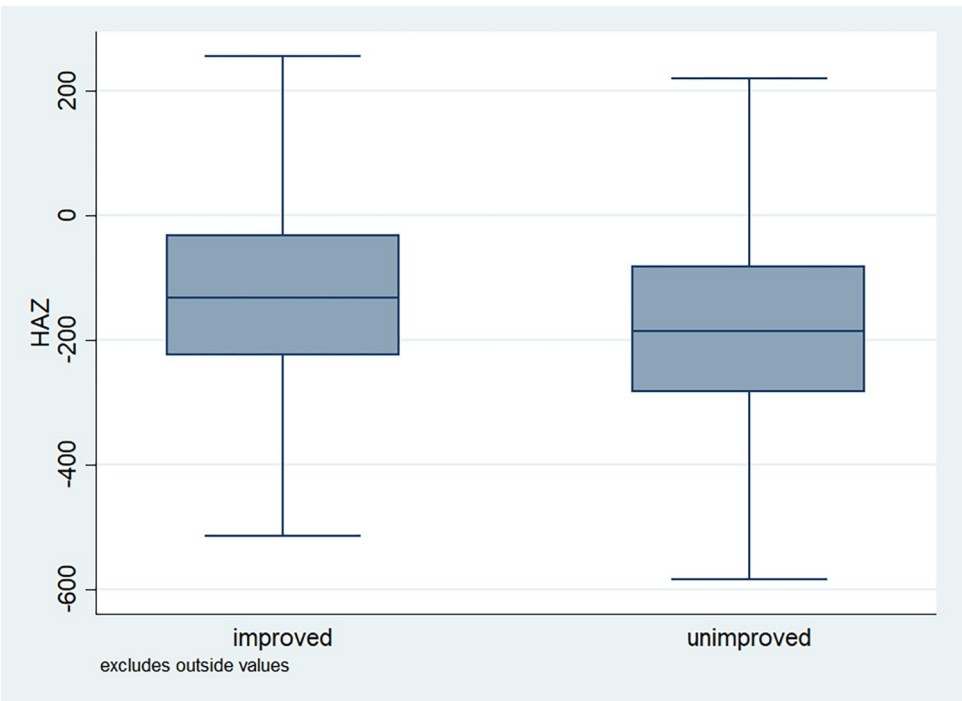

**Fig 9. Box plot of HAZ score by sanitation facility, NFHS, 2015–16, India.**

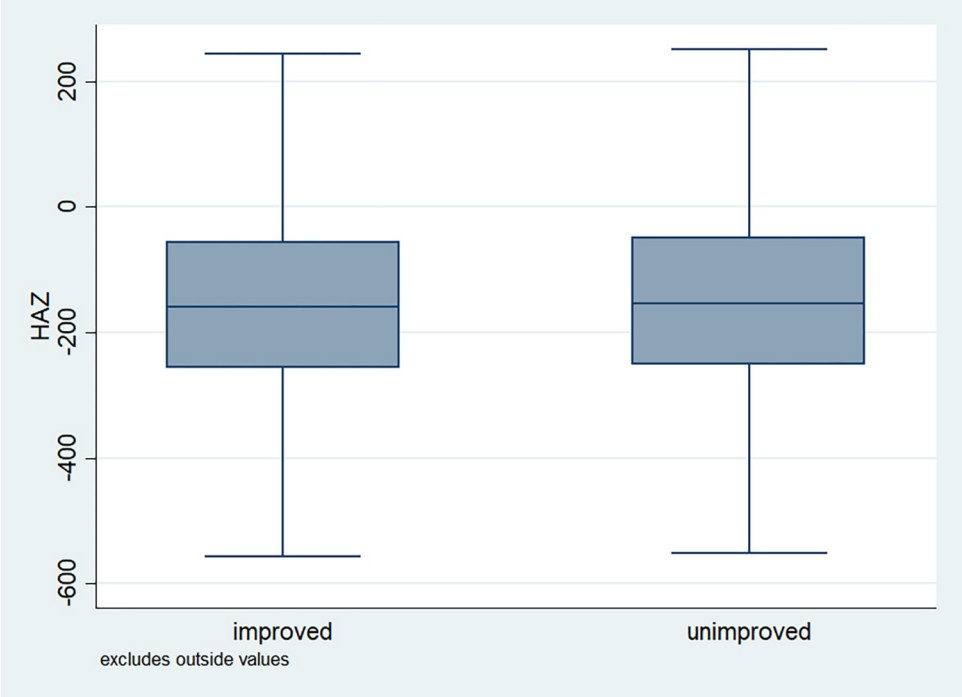

**Fig 10. Box plot of HAZ score by source of drinking water, NFHS, 2015–16, India.**

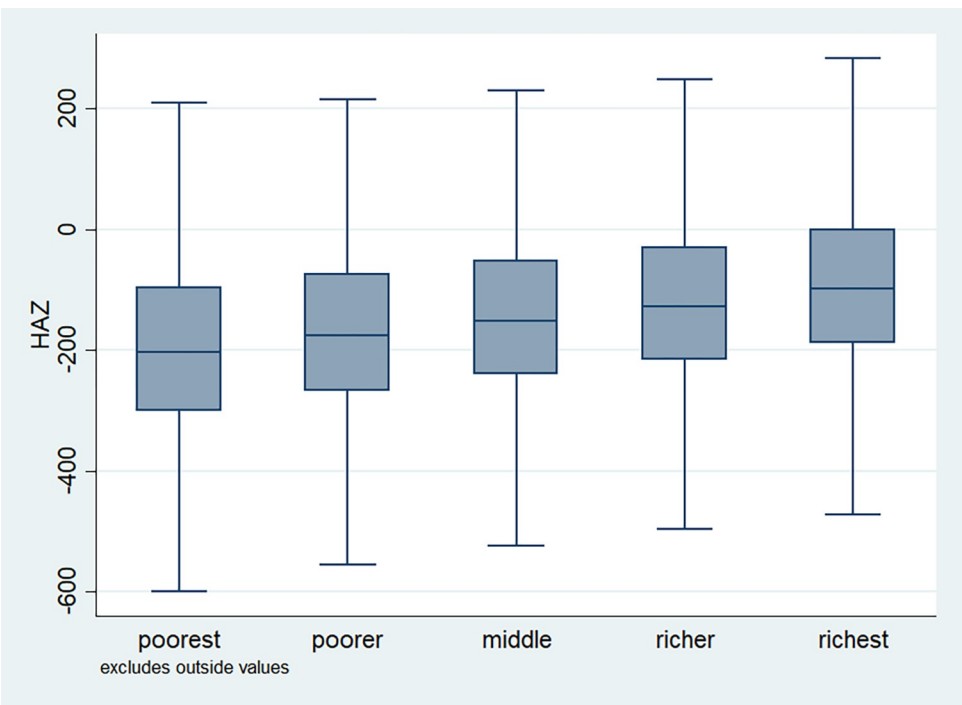

**Fig 11. Box plot of HAZ score by wealth quintile, NFHS, 2015–16, India.**

prevalence than rest of the children and among them children from the poorest wealth quintile showed comparatively higher burden of stunting.

Gender of the child did not show a substantial difference in stunting burden across the wealth quintiles. At the national level, children with no immunization showed higher stunting level (42%) followed by full immunization and partial immunization. Mother's education showed a clear gradient in terms of the stunting prevalence within every class of wealth quintiles. Subject to wealth poverty children of the no educated mothers carried the higher burden of stunting across India. It is observed that children of mothers with no education and from the poorest wealth quintile carried quite high prevalence of stunting (55%). Mother's anthropometry showed a large difference in the burden of child stunting and the prevalence was as high as 57% among the children of the stunted mothers (height less than 145 cm). With a sharp difference within wealth quintiles, it was observed that the burden of child stunting was double among the children of stunted mothers from the richest quintile than their counterpart from the same quintile. Though there was a rural-urban gap evident at the national level but within wealth quintiles, no steep gap had been observed across the wealth quintiles.

Access to improved sanitation showed a varying level of stunting burden across the wealth quintiles and access to improved sanitation did show a lower burden of stunting in different wealth quintiles and nationally. Access to safe drinking water did not show much variation in the stunting prevalence. Regionally, the wealth pattern of child stunting was not very uniform although the poorest wealth quintile from all the regions showed a very high burden of stunting with the Central region carrying a prevalence of 54%.

Table 3 presents the prevalence of stunting among under-five children in India by wealth quintile and various background characteristics for the period 2019–21. Stunting prevalence generally decreases as wealth increases, with 43.7% of children in the poorest quintile being stunted compared to 23.4% in the richest quintile. Age is a significant factor, with children

**Table 2. Prevalence (%) of stunting among under-five children by wealth quintile and background characteristics, India, 2015–16.**

| Variables | Poorest (N) | Poorer (N) | Middle (N) | Richer (N) | Richest (N) | Total (N) |
|---|---|---|---|---|---|---|
| **Age in months** | | | | | | |
| 0–5 | 23.8(4441) | 21.8(4210) | 19.3(3329) | 18.8(2700) | 14.8(2262) | 20.3(16942) |
| 6–11 | 32.1(5362) | 25.9(5071) | 21.6(4350) | 17.9(3611) | 14.2(2970) | 23.4(21364) |
| 12–23 | 55.2(10851) | 48.4(9861) | 41.0(8781) | 34.3(7342) | 26.7(5898) | 42.8(42733) |
| 24–35 | 58.2(10906) | 49.5(10132) | 41.3(8701) | 31.7(7339) | 24.2(5824) | 42.9(42902) |
| 36–47 | 59.3(12117) | 49.2(10809) | 40.8(9011) | 30.9(7456) | 24.8(6068) | 43.3(45461) |
| 48–59 | 54.6(11996) | 46.0(10272) | 37.9(8571) | 29.8(7158) | 20.1(5737) | 40.1(43734) |
| **Sex of the child** | | | | | | |
| Male | 52.3(28382) | 44.4(25810) | 37.2(22330) | 30.2(18401) | 23.6(15388) | 39.2(110311) |
| Female | 51.4(27291) | 43.3(24545) | 36.3(20413) | 28.5(17205) | 20.8(13371) | 38.2(102825) |
| **Birth order** | | | | | | |
| 1 | 49.1(13976) | 40.3(16683) | 33.6(16467) | 26.8(15507) | 20.5(14196) | 33.5(76829) |
| 2–3 | 50.7(25577) | 44.3(24005) | 37.9(20938) | 30.7(17189) | 23.8(13349) | 39.2(101058) |
| 4–5 | 54.8(11412) | 48.9(7189) | 43.2(4217) | 37.9(2432) | 28.1(1085) | 49.0(26335) |
| 6+ | 59.7(4708) | 51.9(2478) | 45.3(1121) | 38.1(478) | 36.3(129) | 54.9(8914) |
| **Immunization** | | | | | | |
| No | 51.5(8136) | 44.4(5393) | 37.5(3419) | 30.9(2204) | 24.5(1220) | 43.1(20372) |
| Partial | 49.1(24040) | 40.7(20591) | 34.6(16280) | 28.8(13114) | 21.1(9757) | 37.3(83782) |
| Full | 54.7(23497) | 46.3(24371) | 38.3(23044) | 29.6(20288) | 22.9(17782) | 39.1(108982) |
| **Mother's Education** | | | | | | |
| No education | 55.3(34642) | 49.5(17758) | 44.0(8797) | 39.5(4033) | 38.0(1213) | 50.9(66443) |
| Primary completed | 49.5(9505) | 45.1(9895) | 40.8(6665) | 35.7(3679) | 30.5(1310) | 43.6(31054) |
| Secondary Completed | 43.6(11221) | 39.2(21672) | 34.2(25004) | 28.4(23114) | 23.7(15415) | 33.0(96426) |
| Higher | 40.5(305) | 31.1(1030) | 25.9(2277) | 22.0(4780) | 18.1(10821) | 20.9(19213) |
| **Mother's height** | | | | | | |
| <145cm | 64.9(9809) | 58.5(6844) | 52.2(4372) | 45.9(2628) | 39.9(1285) | 57.2(24938) |
| > = 145 cm | 49.0(45864) | 41.4(43511) | 34.9(38371) | 27.9(32978) | 21.4(27474) | 36.2(188198) |
| **Residence** | | | | | | |
| Rural | 52.0(53208) | 43.7(45031) | 36.1(33069) | 28.5(20250) | 21.9(10967) | 41.6(162525) |
| Urban | 50.2(2465) | 44.7(5324) | 38.7(9674) | 30.3(15356) | 22.5(17792) | 31.3(50611) |
| **Sanitation** | | | | | | |
| Improved | 47.0(4668) | 41.1(16883) | 35.9(26183) | 29.0(31093) | 22.4(28122) | 30.9(106949) |
| Unimproved | 52.3(51005) | 45.0(33472) | 38.0(16560) | 32.0(4513) | 20.3(637) | 46.4(106187) |
| **Drinking water** | | | | | | |
| Improved | 52.3(46967) | 44.2(43049) | 37.1(37554) | 29.4(32226) | 22.4(26768) | 38.9(186564) |
| Unimproved | 48.6(8706) | 39.9(7306) | 34.0(5189) | 29.6(3380) | 21.5(1991) | 36.8(26572) |
| **Regions** | | | | | | |
| North | 50.5(4357) | 43.5(7462) | 37.3(8654) | 31.6(8791) | 24.5(11052) | 34.6(40316) |
| Central | 54.3(20306) | 48.5(14654) | 42.3(10364) | 35.8(8480) | 25.3(7432) | 44.9(61236) |
| East | 51.4(21987) | 41.7(11303) | 33.0(6369) | 24.1(3737) | 17.5(1592) | 42.4(44988) |
| North East | 46.9(6231) | 37.7(10513) | 28.8(7874) | 19.2(4864) | 16.1(2194) | 35.1(31676) |
| West | 49.7(1832) | 45.0(3000) | 40.0(3534) | 30.8(3382) | 22.7(2684) | 35.8(14432) |
| South | 43.9(960) | 40.4(3423) | 33.6(5948) | 26.2(6352) | 18.5(3805) | 29.7(20488) |
| **India** | **51.9(55673)** | **43.8(50355)** | **36.8(42743)** | **29.4(35606)** | **22.3(28759)** | **38.7(213136)** |

aged 12–23 months having the highest stunting prevalence across all wealth groups (39.9% overall), and the disparity between wealth groups is pronounced in this age category. Male children experience higher rates of stunting (36.2%) compared to females (32.6%), and stunting increases with birth order, particularly among children with higher birth orders (44.7% for birth order 6+).

Additionally, stunting is more prevalent among children with unimmunized or partially immunized statuses, and mothers with no education or lower height (<145 cm) also show higher stunting rates. Rural areas (36.1%) and households with unimproved sanitation (42.3%) report higher stunting compared to urban areas (29.9%) and those with improved sanitation (31.6%). Regionally, stunting is highest in the East (37.6%) and Central (36.1%) regions, with the lowest rates observed in the North (28.5%) and South (29.2%). These findings highlight significant socioeconomic and regional inequalities in stunting prevalence in India.

## Concentration of stunting among heterogeneous child population

Table 4 presents the estimates of concentration indices measuring the wealth-based concentration of stunting by background characteristics of the children. This provided a detailed understanding of wealth-based inequality in child nutrition, i.e., the occurrence of stunting among the heterogeneous child population subject to wealth poverty. We observed statistically significant concentration indices among the different sub-population of children. It was found that concentration of the stunted children was higher among the less wealthy within the sub-population. By age of the child, birth order of the child, immunization status, mother's education, mother's anthropometry (height), access to safe drinking water the variation in inequality was large. Among the different age groups of children, children of age 36–47 months showed the largest pro-poor inequality (ECI: -0.28; 95% CI: -.294, -.265) followed by those children in the 24–35 months and 48–59 months. Higher concentration of stunting was observed among the higher age group of children across India. Though there was no much difference observed by gender but the inequality was quite high among both male and female children. Birth order being a significant predictor of child nutrition [48–50], analysing the wealth inequality among the children of different birth orders, it was found that children of first birth order (ECI: -.217; 95% CI: -.228, -.206) showed the highest concentration of stunting and it was lowest among six and higher ordered births. Children's immunization status also demonstrated a varying level of concentration of stunting. Due to wealth poverty the concentration of stunting was found lowest among the fully immunised children (ECI: -.254; 95% CI: -.264, -.244). Mother's educational attainment did show a varying concentration of stunting prevalence and children of higher educated mothers carried a lower concentration of stunting prevalence within them. Mother's anthropometric status- whether stunted or not showed a difference in the concentration of stunted children and with the exposure to lower wealth the concentration of stunted children was observed more among those children whose mother were not stunted. Access to improved sanitation and safe drinking water showed higher concentration of stunted children when exposed to wealth poverty. The regional pattern of concentration indices showed that Eastern region of India carried highest concentration (-.215) of stunted children followed by North East India, Central India and Western India.

Table 5 also provides the Erreygers corrected concentration indices to measure socioeconomic inequalities in child stunting across various background characteristics in India from 2019–21. Across age groups, stunting inequality worsens with increasing age, with the highest concentration of stunting observed in children aged 24+ months (index value: -0.218), indicating severe inequality in stunting at older ages. There is also a significant disparity in stunting based on immunization status, with children who are fully immunized showing greater

**Table 3. Prevalence (%) of stunting among under-five children by wealth quintile and background characteristics, India, 2019–21.**

| Variables | Poorest (N) | Poorer (N) | Middle (N) | Richer (N) | Richest (N) | Total (N) |
|---|---|---|---|---|---|---|
| **Age in months** | | | | | | |
| 0–5 | 26.3(4272) | 26.8(3489) | 25.3(2875) | 22.9(2643) | 21.2(2109) | 24.8(15388) |
| 6–11 | 31.0(4218) | 27.8(3683) | 23.1(3120) | 20.0(2707) | 18.6(2249) | 24.7(15977) |
| 12–23 | 51.0(8688) | 44.9(7522) | 38.6(6181) | 32.9(5553) | 26.6(4749) | 39.9(32693) |
| 24+ | 50.3(9274) | 43.3(7889) | 38.4(6365) | 29.2(5764) | 23.5(4872) | 38.1(34164) |
| **Sex of the child** | | | | | | |
| Male | 45.9(13484) | 39.5(11586) | 36.1(9521) | 29.3(8602) | 25.9(7287) | 36.2(50480) |
| Female | 41.3(12968) | 37.8(10997) | 31.5(9020) | 26.7(8065) | 20.6(6692) | 32.6(47742) |
| **Birth order** | | | | | | |
| 1 | 41.5(7370) | 36.1(7980) | 30.9(7453) | 26.4(7344) | 21.8(6828) | 30.9(36975) |
| 2–3 | 43.0(12702) | 39.2(11257) | 34.8(9275) | 28.9(8248) | 24.6(6638) | 35.0(48120) |
| 4–5 | 47.0(4746) | 43.0(2704) | 41.3(1522) | 32.9(955) | 28.5(464) | 42.7(10391) |
| 6+ | 48.9(1634) | 44.1(642) | 36.9(291) | 24.4(120) | 32.5(49) | 44.7(2736) |
| **Immunization** | | | | | | |
| No | 43.8(1948) | 37.1(1113) | 30.7(740) | 30.0(567) | 23.3(380) | 35.7(4748) |
| Partial | 39.3(13262) | 36.1(11142) | 31.3(8847) | 26.0(7680) | 21.4(6293) | 31.8(47224) |
| Full | 48.6(11242) | 41.5(10328) | 36.5(8954) | 29.8(8420) | 25.2(7306) | 37.0(46250) |
| **Mother's Education** | | | | | | |
| No education | 48.0(10911) | 43.3(5016) | 40.1(2500) | 33.4(1277) | 30.6(439) | 44.3(20143) |
| Primary completed | 43.8(4839) | 41.4(3487) | 34.9(2065) | 36.0(1177) | 27.1(483) | 39.8(12051) |
| Secondary Completed | 39.3(10245) | 36.9(12828) | 33.4(11775) | 27.9(10308) | 26.2(6634) | 33.0(51790) |
| Higher | 30.0(457) | 29.6(1252) | 27.8(2201) | 24.0(3905) | 19.9(6423) | 23.1(14238) |
| **Mother's height** | | | | | | |
| <145cm | 55.9(4620) | 52.3(3139) | 47.7(1917) | 41.8(1291) | 38.9(752) | 50.5(11719) |
| > = 145 cm | 40.9(21832) | 36.2(19444) | 32.1(16624) | 26.7(15376) | 22.4(13227) | 32.2(86503) |
| **Residence** | | | | | | |
| Rural | 43.7(25655) | 38.4(20861) | 32.9(15182) | 27.4(10918) | 21.4(6169) | 36.1(78785) |
| Urban | 43.8(797) | 41.1(1722) | 37.0(3359) | 28.8(5749) | 24.4(7810) | 29.9(19437) |
| **Sanitation** | | | | | | |
| Improved | 41.4(11751) | 37.9(15612) | 33.2(15791) | 27.8(15913) | 23.2(13814) | 31.6(72881) |
| Unimproved | 45.1(14701) | 40.3(6971) | 37.1(2750) | 31.2(754) | 39.1(165) | 42.3(25341) |
| **Drinking water** | | | | | | |
| Improved | 44.0(22095) | 39.0(19977) | 34.3(16546) | 27.9(14957) | 23.5(12737) | 34.8(86312) |
| Unimproved | 40.7(4357) | 35.2(2606) | 29.6(1995) | 28.3(1710) | 21.9(1242) | 31.6(11910) |
| **Regions** | | | | | | |
| North | 37.9(1614) | 36.8(2887) | 32.2(3541) | 28.3(4517) | 21.4(6020) | 28.5(18579) |
| Central | 43.8(7902) | 39.6(6328) | 34.7(4565) | 29.6(3756) | 24.2(3069) | 36.1(25620) |
| East | 43.9(9532) | 37.8(5226) | 32.1(2851) | 23.3(1728) | 20.3(716) | 37.6(20053) |
| North East | 38.8(5791) | 34.7(5015) | 28.8(2898) | 22.4(1524) | 17.1(497) | 33.9(15725) |
| West | 48.9(1176) | 43.5(1770) | 38.7(2150) | 30.1(2269) | 29.0(1715) | 35.7(9080) |
| South | 50.6(437) | 38.0(1357) | 32.7(2536) | 27.5(2873) | 20.9(1962) | 29.2(9165) |
| **India** | **43.7(26452)** | **38.7(22583)** | **33.8(18541)** | **28.0(16667)** | **23.4(13979)** | **34.5(98222)** |

inequality (-0.189) compared to those partially immunized (-0.145). Regarding sex, both males and females experience similar levels of inequality, with slightly higher concentration indices for females (-0.166). Stunting is more concentrated among poorer households, particularly in rural areas (-0.155) compared to urban (-0.132). Notably, stunting inequality is greater among

**Table 4. Concentration index of child stunting by background characteristics, India, 2015–16.**

| Variables | Index value | 95% CI | | Robust std. error | p-value |
|---|---|---|---|---|---|
| *Age in months* | | | | | |
| 0–5 | -0.058 | -0.077 | -0.039 | 0.010 | <0.01 |
| 6–11 | -0.135 | -0.152 | -0.118 | 0.009 | <0.01 |
| 12–23 | -0.221 | -0.236 | -0.205 | 0.008 | <0.01 |
| 24–35 | -0.269 | -0.284 | -0.254 | 0.008 | <0.01 |
| 36–47 | -0.280 | -0.294 | -0.265 | 0.007 | <0.01 |
| 48–59 | -0.266 | -0.280 | -0.252 | 0.007 | <0.01 |
| **Gender of the child** | | | | | |
| Male | -0.225 | -0.234 | -0.215 | 0.005 | <0.01 |
| Female | -0.235 | -0.245 | -0.225 | 0.005 | <0.01 |
| **Birth order** | | | | | |
| 1 | -0.217 | -0.228 | -0.206 | 0.006 | <0.01 |
| 2–3 | -0.204 | -0.214 | -0.194 | 0.005 | <0.01 |
| 4–5 | -0.144 | -0.162 | -0.126 | 0.009 | <0.01 |
| 6+ | -0.132 | -0.160 | -0.103 | 0.014 | <0.01 |
| **Immunization** | | | | | |
| No | -0.193 | -0.216 | -0.171 | 0.011 | <0.01 |
| Partial | -0.208 | -0.218 | -0.198 | 0.005 | <0.01 |
| Full | -0.254 | -0.264 | -0.244 | 0.005 | <0.01 |
| **Mother's Education** | | | | | |
| No education | -0.107 | -0.118 | -0.096 | 0.006 | <0.01 |
| Primary completed | -0.114 | -0.133 | -0.095 | 0.010 | <0.01 |
| Secondary Completed | -0.141 | -0.152 | -0.131 | 0.005 | <0.01 |
| Higher | -0.078 | -0.096 | -0.060 | 0.009 | <0.01 |
| **Mother's height** | | | | | |
| <145cm | -0.168 | -0.190 | -0.145 | 0.012 | <0.01 |
| > = 145 cm | -0.214 | -0.222 | -0.207 | 0.004 | <0.01 |
| **Residence** | | | | | |
| Rural | -0.207 | -0.214 | -0.199 | 0.004 | <0.01 |
| Urban | -0.182 | -0.198 | -0.166 | 0.008 | <0.01 |
| **Sanitation** | | | | | |
| Improved | -0.158 | -0.169 | -0.147 | 0.006 | <0.01 |
| Unimproved | -0.139 | -0.149 | -0.129 | 0.005 | <0.01 |
| **Drinking water** | | | | | |
| Improved | -0.235 | -0.243 | -0.227 | 0.004 | <0.01 |
| Unimproved | -0.202 | -0.224 | -0.180 | 0.011 | <0.01 |
| *Regions* | | | | | |
| North | -0.190 | -0.206 | -0.174 | 0.008 | <0.01 |
| Central | -0.203 | -0.214 | -0.192 | 0.006 | <0.01 |
| East | -0.215 | -0.228 | -0.202 | 0.007 | <0.01 |
| North East | -0.212 | -0.232 | -0.193 | 0.010 | <0.01 |
| West | -0.203 | -0.229 | -0.176 | 0.014 | <0.01 |
| South | -0.168 | -0.188 | -0.148 | 0.010 | <0.01 |
| **India** | **-0.229** | **-0.237** | **-0.222** | **0.004** | <0.01 |

**Table 5. Concentration index of child stunting by background characteristics, India, 2019–21.**

| Variables | No. of obs. | Index value | 95% CI | | Robust std. error | p-value |
|---|---|---|---|---|---|---|
| *Age in months* | | | | | | |
| 0–5 | 15,388 | -0.041 | -0.066 | -0.017 | 0.012 | <0.01 |
| 6–11 | 15,977 | -0.107 | -0.129 | -0.084 | 0.011 | <0.01 |
| 12–23 | 32,693 | -0.196 | -0.212 | -0.179 | 0.009 | <0.01 |
| 24+ | 34,164 | -0.218 | -0.235 | -0.201 | 0.008 | <0.01 |
| **Sex of the child** | | | | | | |
| Male | 50,480 | -0.163 | -0.177 | -0.149 | 0.007 | <0.01 |
| Female | 47,742 | -0.166 | -0.180 | -0.153 | 0.007 | <0.01 |
| **Birth order** | | | | | | |
| 1 | 36,975 | -0.156 | -0.172 | -0.140 | 0.008 | <0.01 |
| 2–3 | 48,120 | -0.149 | -0.163 | -0.135 | 0.007 | <0.01 |
| 4–5 | 10,391 | -0.106 | -0.135 | -0.077 | 0.015 | <0.01 |
| 6+ | 2,736 | -0.120 | -0.169 | -0.070 | 0.025 | <0.01 |
| **Immunization** | | | | | | |
| No | 4,748 | -0.159 | -0.212 | -0.105 | 0.027 | <0.01 |
| Partial | 47,224 | -0.145 | -0.158 | -0.131 | 0.007 | <0.01 |
| Full | 46,250 | -0.189 | -0.203 | -0.175 | 0.007 | <0.01 |
| *Mother's Education* | | | | | | |
| No education | 20,143 | -0.095 | -0.114 | -0.075 | 0.010 | <0.01 |
| Primary completed | 12,051 | -0.091 | -0.119 | -0.064 | 0.014 | <0.01 |
| Secondary Completed | 51,790 | -0.107 | -0.121 | -0.093 | 0.007 | <0.01 |
| Higher | 14,238 | -0.076 | -0.097 | -0.054 | 0.011 | <0.01 |
| **Mother's height** | | | | | | |
| <145cm | 11,719 | -0.123 | -0.155 | -0.092 | 0.016 | <0.01 |
| > = 145 cm | 86,503 | -0.150 | -0.160 | -0.139 | 0.005 | <0.01 |
| *Residence* | | | | | | |
| Rural | 78,785 | -0.155 | -0.165 | -0.144 | 0.005 | <0.01 |
| Urban | 19,437 | -0.132 | -0.156 | -0.107 | 0.012 | <0.01 |
| **Sanitation** | | | | | | |
| Improved | 72,881 | -0.143 | -0.155 | -0.131 | 0.006 | <0.01 |
| Unimproved | 25,341 | -0.072 | -0.090 | -0.053 | 0.009 | <0.01 |
| **Drinking water** | | | | | | |
| Improved | 86,312 | -0.166 | -0.177 | -0.155 | 0.005 | <0.01 |
| Unimproved | 11,910 | -0.146 | -0.175 | -0.117 | 0.015 | <0.01 |
| *Regions* | | | | | | |
| North | 18,579 | -0.135 | -0.154 | -0.116 | 0.010 | <0.01 |
| Central | 25,620 | -0.152 | -0.169 | -0.136 | 0.008 | <0.01 |
| East | 20,053 | -0.152 | -0.171 | -0.134 | 0.010 | <0.01 |
| North East | 15,725 | -0.118 | -0.145 | -0.090 | 0.014 | <0.01 |
| West | 9,080 | -0.149 | -0.189 | -0.110 | 0.020 | <0.01 |
| South | 9,165 | -0.147 | -0.175 | -0.118 | 0.015 | <0.01 |
| **India** | **98,222** | **-0.165** | **-0.175** | **-0.154** | **0.005** | <0.01 |

children of shorter mothers (<145 cm, index value: -0.123) and in regions like Central (-0.152) and East (-0.152), reflecting significant regional disparities. Overall, the national index of -0.165 suggests substantial socioeconomic inequality in child stunting in India, with poorer children disproportionately affected.

## Decomposition of the concentration indices

Results of decomposition of the concentration indices are shown in Tables 6 & 7. The Erreygers concentration index for a category of a particular variable gave the pro-rich/pro-poor distribution of stunting within that subgroup of children subject to wealth distribution and decomposing the concentration indices, we estimated the elasticity, absolute contributions and percentage of contribution in the overall inequality by each of the background characteristics of the children. The measure of elasticity in this case gave the sensitivity to occurrence of stunting within a subgroup of children from the reference group subject to wealth poverty. And the sign indicates the likelihood to stunting among the children with a certain characteristic. In this study we found the estimated concentration indices to be fairly negative suggesting a concentrated distribution of stunting among the children of a certain population characteristics with lesser wealth.

Factors like child age, birth order and sanitation showed positive elasticity with the CIs being negative in some cases. All the other determinants of child stunting showed a negative value. Among the children of different age groups, children in the age group of 12–23, 24–35 and 36–47 months showed the highest elasticity within the range of 0.049–0.051. This showed that children of these age groups were more likely to be stunted compared the reference group of children subject to lesser wealth. The absolute contribution of wealth poverty on stunting was found to be negative for the last two age groups of children. For the rest of the children, the positive absolute contribution suggested that unequal distribution of stunted children in these age groups will decrease if wealth is uniformly distributed. Though the absolute contribution of gender is almost negligible yet the corresponding elasticity was found negative for the female child compared the male child. This indicated that due to wealth inequality, female children were less likely to be stunted than the male child. The respective elasticity for the children of different birth orders had been found positive which indicated that due to wealth poverty higher ordered births were more likely to be stunted than the first ordered births. Immunization status of the children did show a varying burden of stunting and with a negative elasticity (-0.005), fully immunized children were less likely to be stunted. All the categories of mother's education showed negative elasticity suggesting lower likelihood of stunting among children compared the reference group of children. Mother's education as a determinant of child stunting solely explains 33% of the overall inequality due to uneven distribution of wealth. In other way educational attainment shows a pro-rich distribution signifying the fact that mothers from the wealthy households are higher educated and children of higher educated mothers carry the lower prevalence of stunting. Mother's anthropometric status also explained almost 8% of the overall inequality in child stunting given the wealth-based poverty prevailing in India. And child whose mother are not stunted are less likely to be stunted due to pro-poor distribution of wealth poverty. Urban children were less likely to be stunted than the rural children and place of residence as a covariate explained 5% of the overall inequality. Children from those households with no access to improved sanitation showed more likelihood to stunting compared to those who had access to improved sanitation. Sanitation as a factor explained 24% of the overall inequality in stunting among Indian children. Whereas drinking water did not show any absolute contribution to the overall inequality. Among the different regions of India, North-East and Southern part of India show a negative elasticity which suggested that children from these two regions were less likely to be stunted than those children from the Northern part of India. Regional classification of the children thus explained a total of 6% of the overall inequality.

The decomposition of the concentration index of stunting in 2019–21 reveals key disparities across various socioeconomic and demographic factors. Age plays a crucial role, with

**Table 6. Decomposition of concentration indices of stunting by background characteristics, India, 2015–16.**

| Variables | Stunting | | | |
|---|---|---|---|---|
| | **Elasticity** | **CI** | **Absolute contribution** | **% Contribution** |
| *Age in months* | | | | |
| 0–5 | base | base | base | base |
| 6–11 | 0.004 | 0.005 | 0.000 | -0.038 |
| 12–23 | 0.050 | 0.004 | 0.001 | -0.317 |
| 24–35 | 0.049 | 0.011 | 0.002 | -0.900 |
| 36–47 | 0.051 | -0.005 | -0.001 | 0.415 |
| 48–59 | 0.041 | -0.010 | -0.002 | 0.729 |
| Total | | | | **-0.111** |
| **Gender of the child** | | | | |
| Male | base | base | base | base |
| Female | -0.007 | -0.009 | 0.000 | -0.105 |
| Total | | | **-0.105** | |
| **Birth order** | | | | |
| 1 | base | base | base | base |
| 2–3 | 0.007 | 0.004 | 0.000 | -0.042 |
| 4–5 | 0.001 | -0.309 | -0.002 | 0.684 |
| 6+ | 0.001 | -0.450 | -0.002 | 0.893 |
| Total | | | | **1.535** |
| **Immunization** | | | | |
| No | base | base | base | base |
| Partial | 0.001 | -0.032 | 0.000 | 0.051 |
| Full | -0.005 | 0.063 | -0.001 | 0.557 |
| Total | | | | **0.608** |
| **Mother's Education** | | | | |
| No education | base | base | base | base |
| Primary completed | -0.005 | -0.173 | 0.004 | -1.598 |
| Secondary Completed | -0.044 | 0.174 | -0.031 | 13.421 |
| Higher | -0.020 | 0.608 | -0.048 | 20.910 |
| Total | | | | **32.733** |
| **Mother's height** | | | | |
| <145cm | base | base | base | base |
| > = 145 cm | -0.144 | 0.031 | -0.018 | 7.645 |
| Total | | | | **7.645** |
| *Residence* | | | | |
| Rural | base | base | base | base |
| Urban | -0.006 | 0.446 | -0.010 | 4.498 |
| Total | | | | **4.498** |
| **Sanitation** | | | | |
| Improved | base | base | base | base |
| Unimproved | 0.037 | -0.372 | -0.055 | 23.760 |
| Total | | | | **23.760** |
| **Drinking water** | | | | |
| Improved | base | base | base | base |
| Unimproved | -0.002 | -0.019 | 0.000 | -0.049 |
| Total | | | | **-0.049** |
| **Regions** | | | | |

(*Continued*)

**Table 6.** (Continued)

| Variables | Stunting | | | |
|---|---|---|---|---|
| | Elasticity | CI | Absolute contribution | % Contribution |
| North | base | base | base | base |
| Central | 0.013 | -0.121 | -0.006 | 2.690 |
| East | 0.003 | -0.317 | -0.004 | 1.717 |
| North East | -0.001 | -0.172 | 0.000 | -0.171 |
| West | 0.003 | 0.224 | 0.002 | -1.042 |
| South | -0.005 | 0.303 | -0.006 | 2.729 |
| **Total** | | | | **5.924** |

children aged "24 months and above" contributing the most to socioeconomic inequality in stunting (0.379%). This indicates that older children, particularly those in lower socioeconomic groups, are more affected by stunting. The contribution of younger age groups is smaller, suggesting that stunting inequality is less pronounced in infancy. Gender differences also emerge, with female children showing a negative contribution (-0.1719%), indicating that stunting is more concentrated among females in lower socioeconomic groups compared to males. Additionally, higher birth order, especially for families with 4–5 children, is a significant driver of inequality, contributing 4.68% to the overall stunting disparity.

Education, particularly the mother's educational level, also plays a prominent role in influencing stunting inequality. Mothers with higher education contribute significantly to reducing inequality (17.63%), demonstrating that better-educated mothers are more likely to have children who are less stunted, especially in wealthier quintiles. In contrast, mothers with no or primary education have smaller contributions, highlighting the protective effect of education on child health. Other important factors include unimproved sanitation and urban residence, which also contribute negatively to inequality, suggesting that children living in poorer environments with inadequate sanitation are more likely to suffer from stunting. These findings emphasize the need for targeted interventions in the areas of maternal education, birth order, and living conditions to reduce the socioeconomic disparity in child stunting.

## Discussion

This study assessed the wealth-based inequality in child stunting using Erreygers corrected concentration indices (CIs). Additionally, decomposed the CIs to estimate the elasticity, absolute contribution and percentage of contribution for each of the background characteristics used to define the sub-populations of the children. The results demonstrate the inequality in child's nutritional status that prevails across the sub-population of the children considering the global accountability of wealth inequality in India. As per the recent estimates by National Family Health Survey, 2015–16 it is quite evident that the children in India carry a persistently higher prevalence of stunting though poverty reduction and reduction in child malnutrition remain in top among the public health agendas in India. Though there are nutrition specific interventions running across the country still poverty differential in child nutrition is acute and children from certain population groups are more vulnerable subject to the household's economic wellbeing [51, 52]. And In India, still two-fifth of all the children under age five is stunted and the prevalence of stunting varies largely across the heterogeneous child population of India.

It is apparent that rich-poor gap in stunting prevalence is quite stark and more than half of the children from the poorest wealth quintile are stunted nationally. And within the poorest

**Table 7. Decomposition of concentration indices of stunting by background characteristics, India, 2019–21.**

| Variables | Stunting | | | |
|---|---|---|---|---|
| | **Elasticity** | **CI** | **Absolute contribution** | **% Contribution** |
| *Age in months* | | | | |
| 0–5 | base | base | base | base |
| 6–11 | 0.0001 | 0.0075 | 0.0000 | -0.0012 |
| 12–23 | 0.0308 | 0.0045 | 0.0006 | -0.3374 |
| 24+ | 0.0847 | -0.0018 | -0.0006 | 0.379 |
| Total | | | | **0.040** |
| **Sex of the child** | | | | |
| Male | base | base | base | base |
| Female | -0.0182 | -0.0039 | 0.0003 | -0.1719 |
| Total | | | | **-0.1719** |
| **Birth order** | | | | |
| 1 | base | base | base | base |
| 2–3 | 0.0118 | -0.0019 | -0.0001 | 0.0533 |
| 4–5 | 0.0065 | -0.2976 | -0.0077 | 4.6832 |
| 6+ | 0.0021 | -0.4383 | -0.0036 | 2.1967 |
| Total | | | | **6.933** |
| **Immunization** | | | | |
| No | base | base | base | base |
| Partial | 0.0007 | -0.0121 | 0 | 0.0211 |
| Full | -0.0025 | 0.0278 | -0.0003 | 0.1684 |
| Total | | | | **0.190** |
| **Mother's Education** | | | | |
| No education | base | base | base | base |
| Primary completed | 0.0022 | -0.238 | -0.0021 | 1.2855 |
| Secondary Completed | 0.0022 | -0.238 | -0.0021 | 1.2855 |
| Higher | -0.0141 | 0.5138 | -0.029 | 17.6278 |
| Total | | | | **20.199** |
| **Mother's height** | | | | |
| <145cm | base | base | base | base |
| > = 145 cm | -0.1328 | 0.0295 | -0.0157 | 9.5398 |
| Total | | | | **9.5398** |
| *Residence* | | | | |
| Rural | base | base | base | base |
| Urban | -0.0065 | 0.443 | -0.0116 | 7.0457 |
| Total | | | | **7.0457** |
| **Sanitation** | | | | |
| Improved | base | base | base | base |
| Unimproved | 0.0174 | -0.4617 | -0.0321 | 19.4905 |
| Total | | | | **19.4905** |
| **Drinking water** | | | | |
| Improved | base | base | base | base |
| Unimproved | -0.002 | 0.031 | -0.0002 | 0.1505 |
| Total | | | | **0.1505** |
| **Regions** | | | | |
| North | base | base | base | base |
| Central | 0.0117 | -0.0775 | -0.0036 | 2.2129 |

*(Continued)*

**Table 7.** (Continued)

| Variables | Stunting | | | |
|---|---|---|---|---|
| | Elasticity | CI | Absolute contribution | % Contribution |
| East | 0.0096 | -0.3151 | -0.0121 | 7.3351 |
| North East | 0.0007 | -0.3134 | -0.0009 | 0.5604 |
| West | 0.0077 | 0.2188 | 0.0068 | -4.1203 |
| South | 0.003 | 0.2899 | 0.0035 | -2.1053 |
| Total | | | | **3.883** |

section, children of stunted mothers carry critically higher burden of stunting which is also consistent with a previous study findings [53]. The prevalence of child stunting is lopsided very high in the lower wealth quintiles from different sub-populations. The concentration indices explained the unequal distribution of stunted children within the population sub-groups across India. The estimate of CI shows a statistically significant and negative concentration index nationally and the pattern remained consistently high and negative among all the subgroups (ranging in between 0.06 to 0.28) suggesting a pro-poor distribution of the stunted children. Like previous studies [48, 54–58], we also found that among the different factors, age of children, birth order, mother's educational attainment, mother's anthropometry (height), place of residence, sanitation (access to improved toilet facility) and the region a child where he/she belongs to are the key factors explaining the gap in stunting subject to the distribution of wealth. Age pattern inequalities relative to nutritional status showed that children of age 36–47 & 48–59 months had relatively high pro poor/rich inequalities. Result for the gender pattern of CI shows no significant differential in stunting prevalence in the presence of persisting wealth inequality. With a negligible absolute contribution in the overall inequality, gender of the child does not show any strong evidence on gender-based inequality in nutrition given the wealth poverty situation in India.

Given the prevailing wealth inequality across India, it is found that the burden of stunting is disproportionate by population characteristics and each of the characters contributes differently to the overall inequality in child stunting. In this regard, mother's educational attainment, sanitation condition, mother's stature and place of residence of the children play a dominant role to contribute to the overall inequality in child malnutrition. It is found that once adjusted for all other variables within the framework, 1 & 2–3 ordered births show higher pro-poor concentration of stunted children than the rest of the higher ordered births. The possible reason could be that a large population (84% of the total children) of lower ordered births (1, 2–3) compared the smaller population (16% of the total children) of higher ordered births is exposed to a pro-poor distribution of wealth and thus the larger population of lower ordered births of children carry the higher concentration of stunted children given the wealth distribution.

The analysis of recent round of the NFHS data also indicates that stunting prevalence among under-five children in India reveals significant wealth-poverty inequalities across different socio-demographic groups. The Erreygers corrected concentration indices consistently show that stunting is concentrated among children from poorer households. For instance, stunting is notably higher among children aged 24 months and above, particularly in lower wealth quintiles. The concentration index for this age group (-0.218) underscores that stunting in older children is disproportionately found among the poorest households. This trend is also evident across gender, where both males and females exhibit high levels of inequality, with concentration indices of -0.163 and -0.166, respectively. Similarly, the decomposition analysis

highlights that factors such as higher birth order, unimproved sanitation, and lack of maternal education contribute significantly to the inequality in stunting. Children from larger families or those with mothers who have lower education levels are disproportionately affected by stunting, and this disparity widens with economic disadvantage.

Wealth-related inequality in stunting is also influenced by geographic and environmental factors, such as residence, sanitation, and access to clean water. Urban children fare slightly better than rural children, although both groups show stunting concentrated in poorer households. Regions such as the East and Central India contribute heavily to stunting inequality, as demonstrated by their high negative concentration indices. Poor sanitation is another critical determinant, with unimproved sanitation contributing almost 20% to stunting inequality. Overall, the data reflect a clear socioeconomic gradient in stunting prevalence, where children from the poorest families, particularly those with low maternal education, high birth order, and poor living conditions, bear the brunt of stunting. Addressing these disparities requires a multifaceted approach that includes improving maternal education, providing better sanitation, and addressing geographic disparities to mitigate wealth-driven inequalities in child nutrition outcomes.

This particular study demonstrated the evidence on wealth-based inequality in stunting prevalence across the heterogeneous child population of India using wealth as a measure of economic wellbeing of the household than income or consumption expenditure of the household. Though across the surveys, the measure of economic wellbeing varies, there are certain merits and demerits associated with each of the measures. Arguably, a direct measurement of household's total income is a good measure of money metric poverty and the purchasing capacity of a household. While the measure of consumption expenditure of a household provides direct information on expenditure incurred on food items as well as the calorie consumed. On the other hand, wealth based measure is quite an indirect measure as well as a proxy measure to economic wellbeing [37, 59]. Like other standard Demographic Health Surveys, NFHS also collected the wealth-based information on an exhaustive range set of asset-based indicators and provides the standardised measure of wealth based economic wellbeing of a household which is extensively used in this study to capture the poverty (wealth) induced nutritional inequality among Indian children.

This study adds to the existing knowledge of nutritional inequality among under-five children and the uneven distribution of stunting burden among the heterogeneous child population subject to the distribution of wealth using the most recent rounds of National Family Health Survey, 2015–16 & 2019–21. In this regard, this study corroborates the most recent pattern of nutritional inequality and identifies the sub-group of children vulnerable to undernutrition with higher concentration of stunted children. Thus, it is global to mention that poverty being a major driver of child undernutrition it needs the most careful attention to alleviate poverty and to make those households food secure from the lowest wealth quintiles to uplift the nutritional health of the children.

In the absence of a direct measure of the household's economic status, we utilised a proxy measure (wealth index) of economic wellbeing to examine the related inequality in child nutrition and propagates the message of heterogeneous concentration of stunted children in the different sub-groups. As the present study shows marked rich-poor differences in child stunting across the sub-populations, it is recommended that poverty alleviation is key to fight child undernutrition in India along with running nutrition-sensitive and nutrition-specific programs. It is known that the food poverty policies across countries mostly target to improve the dietary pattern of the children from the poor families. These food poverty policies addresses children's needs by recommending monetary help to parents, feeding the children and improved access to healthy and affordable diet [21]. In India, the child nutrition policies are

formulated and implemented by the Ministry of Women and Child Development (MWCD). With a special focus to nutrition, the Integrated Child Development Services (ICDS) scheme in India is aimed to reduce the burden of malnutrition and also sensitises the mothers to look after their children's nutritional needs. But still children from the poorer households carry the highest burden of stunting and given the wealth distribution, the prevalence of stunting is disproportionate across the sub-populations. Thus, it is of utmost importance to formulate a responsive policy which can essentially reduce the economic inequality and improve the economic status of the poorer households.

The limitation of this study is that this is a cross-sectional study and this fails to draw a causal inference between wealth poverty and child stunting rather this study is specific to examine the concentration of stunted children across the sub population with a major thrust to identify the related inequality in child stunting due to wealth poverty across India. Potentially, this study estimated the Erreygers corrected concentration indices and decomposed it to measure the elasticity and the associated contribution in the overall inequality of a particular factor which has not been attempted previously.

## Conclusion

Socio-economic inequality in child nutrition is persistent in India given the wealth poverty situation across the subpopulation. Though poverty reduction is one of the SDG goals implemented in India, this study reinforces the importance of poverty alleviation which may help to reduce the overall inequality in child nutrition parallel to improvement in terms of the other health outcomes in the population. It is also found that there are several factors like, mother's education, sanitation condition, mother's anthropometrical height and place of residence plays a strong role in the overall inequality. Thus, careful focus and implementation of appropriate interventions to improve mother's education, sanitation in the rural areas and monitoring mother's nutritional health along with taking care of the diet and food pattern among the young women can reflect on their stature which in turn may help to reduce the prevalence of stunting among children in India.

## Supporting information

**S1 Appendix. Sub-population-specific patterns in the HAZ scores of under-five children in India, NFHS, 2019–21.**
(PDF)

## Author Contributions

**Conceptualization:** Junaid Khan, Sanjay K. Mohanty.

**Data curation:** Junaid Khan.

**Formal analysis:** Junaid Khan.

**Methodology:** Junaid Khan.

**Software:** Junaid Khan.

**Supervision:** Sanjay K. Mohanty.

**Writing – original draft:** Junaid Khan.

**Writing – review & editing:** Sanjay K. Mohanty.

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
