## [Decision Letter · Decision Letter 0]

23 Apr 2021

PONE-D-21-08355

The extent of wealth poverty induced inequality in nutrition among children born during 2010-16 in India

PLOS ONE

Dear Dr. Khan,

Thank you for submitting your manuscript to PLOS ONE. After careful consideration, we feel that it has merit but does not fully meet PLOS ONE’s publication criteria as it currently stands. Therefore, we invite you to submit a revised version of the manuscript that addresses the points raised during the review process.

We look forward to receiving your revised manuscript.

Kind regards,

Kannan Navaneetham, PhD

Academic Editor

PLOS ONE

Journal Requirements:

4. Please amend your manuscript to include your abstract after the title page.

Reviewers' comments:

Reviewer's Responses to Questions

**Comments to the Author**

1. Is the manuscript technically sound, and do the data support the conclusions?

Reviewer #1: Partly

Reviewer #2: No

2. Has the statistical analysis been performed appropriately and rigorously? 

Reviewer #1: Yes

Reviewer #2: No

3. Have the authors made all data underlying the findings in their manuscript fully available?

Reviewer #1: Yes

Reviewer #2: Yes

4. Is the manuscript presented in an intelligible fashion and written in standard English?

Reviewer #1: No

Reviewer #2: No

5. Review Comments to the Author

Reviewer #1: The title is to be simplified as “Poverty induced inequality in nutrition among children born during 2010-16 in India”

Would the authors be considering to use the word “gender” in leu of “sex” throughout the texts?

Authors are requested to ensure consistency in the use of terminology - ……..the economic inequality in ……;…..wealth based inequality…..; …..wealth poverty….;….poverty based inequality…..;……wealth disparity…;and so on…

Abstract section

Present tense form is to be used throughout.

……poverty gap in child nutrition is largest. <– the expression is not clear and also, we need to use “the” before largest.

Please capture urban / rural with place of residence.

The main texts

Overall, the flow of writing the texts including attention to the grammar deserves a thorough revisiting.

The uniqueness of the study deserves a better foundation with an explicit comprehension. Authors need to establish the impact of the study on “Lives and Society” over “Academic excellence” with a candid discussion for translating Science to Policy and Action.

Page3-4: This part of the texts is non-substantive in the context -> India still stands as the development paradox and in terms of overall human development index (HDI) ranks 130 globally though it has made an improvement in terms of absolute reduction in poverty, increase in life expectancy and improvement in education and standard of living but failed in terms of social progress in the last decade [14-17]. However, the inequality in developmental parameters like health, education and health care access are large within the country [18, 19]. At the same time it is also evident that it is the poor with low income show poor health status and malnourishment among

those children from the poor families [20, 21].

Instead, authors can capture potential contribution of this study to the development trajectory of India in the spotlight of Sustainable Development Goals.

Page3:…..the wealth gap in child health….. <– not understood.

Should we say, « wealth related inequality » as is captured in the texts or poverty induced inequality?

Page3: How Gonzalo-Almorox and Urbanos-Garrido method is different from Oaxaca-type decomposition? Please explain in “Methods” section. Is the “Method choice” having the “Power to change views”?

Page4: Authors have used data from the National Family Health Survey (NFHS)-2015-16. Stunting becomes evident during 1st 1000 days of life. How the authors claim that the effect on the children born in 2015/2016 are sufficiently understood to be inferred?

What is the formulation process of the mathematical models used in the study?

The steps followed to present the contribution of the determinants are not found in the texts.

How did the authors derive the percentage contribution of each of the determinants? Please explain in “Methods” section.

Page8: Such a claim –> [socially excluded groups scheduled caste or scheduled tribe.] requires to be supported by citation.

Page9: This hints about the differential of nutritional status among the

children of a specific birth order. How do we know that this could be the only reason?

Page9-10: [...to wealth poverty] <– can it be extent of material affluence?

Page11: [Analysing the wealth inequality among the children of different birth orders, it is found that children of first birth order (ECI: -.217; 95% CI: -.228,-.206) show the highest concentration of stunting and it is lowest among six and higher ordered births.] implying what?

Page12 and throughout the texts: Can we not use a lucid expression in lieu of [higher wealth poverty]?

Page14: what do the authors mean by -> [Chronic poverty situation….]?

Page14: what do the authors mean by –> [among children of different socio-demographic characteristics.]?

Page14: [….the gender pattern shows a diminished differential in stunting prevalence…..] <- not well understood.

Please put the legend appropriate for each figure.

Table1. Can we not use the word percentage distribution in the heading instead of using the word frequency?

Also, authors are requested to cluster the states / union territories by region and appropriately discuss the results from the analysis.

Table2. Please write the measure of prevalence in the header appropriately.

Table3. Please make the header explicit – not comprehendible at all, as it stands now. How have the authors addressed the inadequacy of sample size in the analysis?

Table 7. Does it mean that in 14 out of 36 States / Union Territories the scenario is better? If, so, can the authors bring some insights into the discussion interconnected challenges and systematic effects from the perspectives of systems thinking?

Reviewer #2: Stunting among children is quite an interesting topic especially for India, which is part of the BRICS block. However, I was surprised not to see anything about the COVID-19 shocks on stunting among children. One can not separate malnutrition from food security entirely, of which the pandemic has done a huge impact on food security and exacerbated health inequalities in general. Secondly, the decomposed model could not explain about 24% of the variations as some of the crucial determinants of stunting were left out. Decomposition by states is not meaningful if the underlying characteristics of the states are not known, not sure why the authors decomposed by states. Results were reported in present tense yet they should be reported in the past tense. The discussion was poorly written might need total restructuring as the study results were not related to existing study findings.

6. PLOS authors have the option to publish the peer review history of their article (what does this mean?). If published, this will include your full peer review and any attached files.

Reviewer #1: No

Reviewer #2: No

---

## [Author Response · Author response to Decision Letter 0]

27 Jun 2021

Response to Reviewers

PONE-D-21-08355

The extent of wealth poverty induced inequality in nutrition among children born during 2010-16 in India

PLOS ONE

Reviewer #1: 

Comment1: The title is to be simplified as “Poverty induced inequality in nutrition among children born during 2010-16 in India”

Reply: Thank you very much for your kind suggestion to improve the title. The title has been changed. 

Comment2: Would the authors be considering to use the word “gender” in lieu of “sex” throughout the texts?

Reply: Thank you very much. The word has been replaced throughout the manuscript as per the suggestion. 

Comment3: Authors are requested to ensure consistency in the use of terminology - ……..the economic inequality in ……;…..wealth based inequality…..; …..wealth poverty….;….poverty based inequality…..;……wealth disparity…;and so on…

Reply: Thank you very much for your kind suggestion. A consistency has been maintained while using the terminologies. 

Abstract section

Comment4: Present tense form is to be used throughout.

……poverty gap in child nutrition is largest. <– the expression is not clear and also, we need to use “the” before largest.

Please capture urban / rural with place of residence.

Reply: Thank you very much. The present tense is used in the abstract. The sentence is rephrased. 

The main texts

Overall, the flow of writing the texts including attention to the grammar deserves a thorough revisiting.

Reply: Thank you very much. The document has been revisited taking special care of the grammar.

Comment5: The uniqueness of the study deserves a better foundation with an explicit comprehension. Authors need to establish the impact of the study on “Lives and Society” over “Academic excellence” with a candid discussion for translating Science to Policy and Action.

Reply: The usefulness of this study is discussed taking care of the policy perspectives within the scope of the study. 

Comment6: Page3-4: This part of the texts is non-substantive in the context -> India still stands as the development paradox and in terms of overall human development index (HDI) ranks 130 globally though it has made an improvement in terms of absolute reduction in poverty, increase in life expectancy and improvement in education and standard of living but failed in terms of social progress in the last decade [14-17]. However, the inequality in developmental parameters like health, education and health care access are large within the country [18, 19]. At the same time it is also evident that it is the poor with low income show poor health status and malnourishment among those children from the poor families [20, 21].

Instead, authors can capture potential contribution of this study to the development trajectory of India in the spotlight of Sustainable Development Goals.

Reply: The cited literatures in this paragraph support the scientific content and substantiate the nexus between the present situation of the child health parameter in terms of stunting and India’s development in terms of absolute reduction in poverty, increase in life expectancy and improvement in education and standard of living. Within the scope of the study and to introduce the goal of the study this paragraph gives a brief overview of India’s development trajectory in the context of child stunting. There are many development goals under the SDGs which are targeted to achieve and India’s progress is quite convincing still child undernutrition remained one of the major public health challenges in India and almost two-fifth of the total children under age five in India are still stunted. Within the context and goal of this study, the introduction section gives a detailed snapshot of the problem and the rationale of the study. Thank you very much. 

Comment7: Page3:…..the wealth gap in child health….. <– not understood.

Should we say, « wealth related inequality » as is captured in the texts or poverty induced inequality?

Reply: Thank you very much for the kind suggestion. It has been rephrased accordingly in the revised manuscript. Page No-4, Line No-82

Comment8: Page3: How Gonzalo-Almorox and Urbanos-Garrido method is different from Oaxaca-type decomposition? Please explain in “Methods” section. Is the “Method choice” having the “Power to change views”?

Reply: The Oaxaca-type decomposition is used when the dependent variable is continuous in nature whereas the Gonzalo-Almorox and Urbanos-Garrido method is used when the dependent variable is dichotomous in nature. It has been explained in the methods section. 

No, the decomposition method chosen in this study is apt for the study variable and the estimates are not influenced by the choice of the method. 

Comment9: Page4: Authors have used data from the National Family Health Survey (NFHS)-2015-16. Stunting becomes evident during 1st 1000 days of life. How the authors claim that the effect on the children born in 2015/2016 are sufficiently understood to be inferred?

Reply: The NFHS survey is a cross-sectional survey and the age of the children and the anthropometric measures are measures on the date of interview. Thus this study is limited to measure the effect of first 1000 days of life. At the same time this study is limited to measure the cohort effect except the age fixed effect. To mention, NFHS surveys of different rounds provide the estimates of undernutrition among under five children and the datasets are grossly used the socio-economic and demographic patterns and determinants of different child health parameters including child undernutrition within a cross-sectional framework. Similarly, this study also used the data information from NFHS-4 and examined the inequality in child nutrition using a novel decomposition approach.

Comment10: What is the formulation process of the mathematical models used in the study?

Reply: The methods section provides the details of the equations and the formulas for the statistical analysis that was being adopted in this study.

Comment11: The steps followed to present the contribution of the determinants are not found in the texts. How did the authors derive the percentage contribution of each of the determinants? Please explain in “Methods” section. 

Reply: Thank you very much for your kind suggestion. It has been explained in the methods section. Page-9, Line No-202/208

Comment12: Page8: Such a claim –> [socially excluded groups scheduled caste or scheduled tribe.] requires to be supported by citation.

Reply: Thank you very much for your kind comment. Citations are made in the revised manuscript. 

https://idsn.org/wp-content/uploads/user_folder/pdf/New_files/Key_Issues/MDG_issues/Making_post-2015_matter__OXFAM_India__2013.pdf

https://sustainabledevelopment.un.org/content/documents/11145Social%20exclusion%20and%20Inequality-Study%20by%20GCAP%20India%20.pdf

Comment13: Page9: This hints about the differential of nutritional status among the

children of a specific birth order. How do we know that this could be the only reason?

Reply: The line describes the differential in HAZ scores among the children of different birth orders which is based upon a bivariate box plot analysis of the data. Further the paragraph discusses the varying level of HAZ scores among the children of different population characteristics. As we proceed with the analysis, we discussed the adjusted effects of a particular factor within a multivariate framework. The line has been revised to bring better clarity into the sentence. Page No-10, Line No- 222. 

Comment14: Page9-10: [...to wealth poverty] <– can it be extent of material affluence?

Reply: Like the DHS surveys, NFHS also only provides the wealth based measure of household’s economic wellbeing. This measure is based upon the asset information from the households. Thus it could be said that wealth measure of economic wellbeing captures the material affluence of the household. Thank you very much. 

Comment15: Page11: [Analysing the wealth inequality among the children of different birth orders, it is found that children of first birth order (ECI: -.217; 95% CI: -.228,-.206) show the highest concentration of stunting and it is lowest among six and higher ordered births.] implying what?

Reply: The measure provides the corresponding value associated with the concentration of the stunted children subject to wealth distribution across the subpopulations. As we see, of the total children 84% of them are lower birth order (3 or less) whereas only 4% of them are of 6+ ordered births. Once adjusted all other variables within the framework, 1 & 2-3 ordered births show higher concentration of stunted children than the rest of the higher ordered births. Thus it could be pointed out that a large population of lower ordered births (1, 2-3) compared the smaller population of higher ordered births is exposed to a pro-poor distribution of wealth and thus the population of lower ordered births of children carry the higher concentration of stunted children. We discussed this point in the discussion section.

Comment16: Page12 and throughout the texts: Can we not use a lucid expression in lieu of [higher wealth poverty]?

Reply: Thank you very much for your kind comment. It has been rephrased taking care of the sentence and its meaning. Page No-14, Line No- 346 & 351 

Comment17: Page14: what do the authors mean by -> [Chronic poverty situation….]?

Reply: It has been rephrased. Thank you so much. Page No-15, Line No- 357

Comment18: Page14: what do the authors mean by –> [among children of different socio-demographic characteristics.]?

Reply: Thank you very much for your kind suggestion. The line has been rephrased. Page No- 16, Line No- 376

Comment19: Page14: [….the gender pattern shows a diminished differential in stunting prevalence…..] <- not well understood.

Reply: Thank you very much. It has been rephrased. Page No-16, Line No- 381. 

Comment20: Please put the legend appropriate for each figure.

Reply: Thank you very much for the kind suggestion. It has been put.

Comment21: Table1. Can we not use the word percentage distribution in the heading instead of using the word frequency?

Reply: Thank you so much. We have changed the heading.

Comment22: Also, authors are requested to cluster the states / union territories by region and appropriately discuss the results from the analysis.

Reply: Yes it has been done so. Thank you very much for your kind comment. 

Comment23: Table2. Please write the measure of prevalence in the header appropriately.

Reply: It has been corrected. Thank you so much. 

Comment24: Table3. Please make the header explicit – not comprehendible at all, as it stands now. How have the authors addressed the inadequacy of sample size in the analysis?

Reply: The total analytical sample of the study is 225,002. Though state specific sample by wealth quintile are scanty for few of the states but state level aggregated estimation is most certainly possible as the sample size is sufficient for unbiased estimation of stunting. As per the sample size issue, the state level representation has been dropped to keep the focus of the study unique and one dimensional. So, this table is dropped in the revised manuscript.

Comment25: Table 7. Does it mean that in 14 out of 36 States / Union Territories the scenario is better? If, so, can the authors bring some insights into the discussion interconnected challenges and systematic effects from the perspectives of systems thinking?

Reply: The state level analysis has been dropped to maintain a good focus over the sub-population analysis. Additionally, we would like to mention that sample size is not adequate for multiple states in the different wealth quintile category and thus we decided not to go for the state level analysis. 

Reviewer #2: 

Comment26: Stunting among children is quite an interesting topic especially for India, which is part of the BRICS block. However, I was surprised not to see anything about the COVID-19 shocks on stunting among children. One can not separate malnutrition from food security entirely, of which the pandemic has done a huge impact on food security and exacerbated health inequalities in general. 

Reply: This study is based upon National Family Health Survey, 2015-16 and the study children are those who are under age five prior to the survey. The primary objective of this study was to examine the extent of stunting given the uneven distribution of wealth across the sub-population. While this study applied novel approach to decompose the concentration indices. The onset COVID in India was 2020 February, and NFHS, 2015-16 does not have any information regarding the COVID pandemic. Although COVID pandemic has impacted the children in many different ways starting from nutrition, immunization to health care utilization but this study is limited to NFHS dataset only. On the other hand there is no exhaustive dataset available on child nutrition in the context of COVID for India which could otherwise be utilised to infer at the unit level. All the inferences in this study are drawn at the micro level. Thus this study is limited to understand the COVID 19 shocks on child stunting. 

Comment27: Secondly, the decomposed model could not explain about 24% of the variations as some of the crucial determinants of stunting were left out. Decomposition by states is not meaningful if the underlying characteristics of the states are not known, not sure why the authors decomposed by states. Results were reported in present tense yet they should be reported in the past tense. The discussion was poorly written might need total restructuring as the study results were not related to existing study findings.

Reply: This study did not aim to explore the determinants of stunting among Indian children; rather examined the nutritional status across the different sub-population (socio-economic and demographic strata) given the wealth distribution within a specific sub-population. This is to confirm that, the decomposed model explained almost 77% of the overall inequality (Table 6). Variables like age, sex of the child and drinking water explained a very low and negative variation. 

Although we did not decompose the factor level contribution; rather estimated the concentration indices across the subpopulation and across the state first and then the concentration indices were decomposed. So, the analytical strategy is different in this study from the conventional factor level decomposition. This is to mention that mother’s education, sanitation, drinking water, wealth index are the underlying determinants of child stunting and are essentially considered in this study. As per the suggestion by one of the reviewer, state level decomposition is dropped from this study.

Results are reported in past tense in the revised manuscript. And the discussion section is also revised within the scope of the study framework. We the authors are very thankful for your kind suggestions to help us improve the scientific content of the study and its relevance.

---

## [Decision Letter · Decision Letter 1]

18 Oct 2021

PONE-D-21-08355R1Poverty induced inequality in nutrition among children born during 2010-16 in IndiaPLOS ONE

Dear Dr. Khan,

Thank you for submitting your manuscript to PLOS ONE. After careful consideration, we feel that it has merit but does not fully meet PLOS ONE’s publication criteria as it currently stands. Therefore, we invite you to submit a revised version of the manuscript that addresses the points raised during the review process.

The revised paper was reviewed by the same set of reviewers. One reviewer has suggested that some of the responses can be appended into the manuscript and also raised concerns about the language. Kindly revise the paper accordingly. 

We look forward to receiving your revised manuscript.

Kind regards,

Kannan Navaneetham, PhD

Academic Editor

PLOS ONE

Journal Requirements:

Reviewers' comments:

Reviewer's Responses to Questions

**Comments to the Author**

1. If the authors have adequately addressed your comments raised in a previous round of review and you feel that this manuscript is now acceptable for publication, you may indicate that here to bypass the “Comments to the Author” section, enter your conflict of interest statement in the “Confidential to Editor” section, and submit your "Accept" recommendation.

Reviewer #1: (No Response)

Reviewer #2: All comments have been addressed

2. Is the manuscript technically sound, and do the data support the conclusions?

Reviewer #1: Yes

Reviewer #2: Yes

3. Has the statistical analysis been performed appropriately and rigorously? 

Reviewer #1: Yes

Reviewer #2: Yes

4. Have the authors made all data underlying the findings in their manuscript fully available?

Reviewer #1: Yes

Reviewer #2: Yes

5. Is the manuscript presented in an intelligible fashion and written in standard English?

Reviewer #1: No

Reviewer #2: Yes

6. Review Comments to the Author

Reviewer #1: Abstract section

Reply: Thank you very much. The present tense is used in the abstract. The sentence is rephrased.

This is not an honest statement –

Page2: Line 8-> [… … … this paper examined… … …].

Page2: Lines19-26-> [About … … … place of residence (5%).].

Page2: Lines30-31-> [Mother’s education… … … across India.].

Reply: Thank you very much. The document has been revisited taking special care of the grammar.

Yes, to some extent but far below the acceptable level.

Highlights (not an exhaustive list) of the incorrect response

Page9: Line 199-> [As we utilized … … …], the word “utilized” is not the apt word here.

Page9: Lines205-207-> [The estimated partial effects from the probit model are used to compute the contributions of the explanatory variables considered in the study framework. In summary, the factor level contributions are calculated… … …] why present tense form in Method section and that too inconsistent use of tense in the section?

Page 10: Line 222-> [Mothers to… … …]?

Page15: Line354 and Page17: Line407-> All on a sudden expanded form of NFHS [National Family Health Survey] appears.

Results and Discussion section(s) are to be in the “past tense” form but the texts do not follow any consistent pattern of tense use.

Page15: Lines350-352-> Comprehension with no English structure – [Additionally, decomposed the CIs to estimate the elasticity, absolute contribution and percentage of contribution for each of the background characteristics used to define the sub-populations of the children.].

Reply: The usefulness of this study is discussed taking care of the policy perspectives within the scope of the study.

Misleading response; not found in the revised submission.

Reply: The cited literatures in this paragraph support the scientific content and substantiate the nexus between the present situation of the child health parameter in terms of stunting and India’s development in terms of absolute reduction in poverty, increase in life expectancy and improvement in education and standard of living. Within the scope of the study and to introduce the goal of the study this paragraph gives a brief overview of India’s development trajectory in the context of child stunting. There are many development goals under the SDGs which are targeted to achieve and India’s progress is quite convincing still child undernutrition remained one of the major public health challenges in India and almost two-fifth of the total children under age five in India are still stunted. Within the context and goal of this study, the introduction section gives a detailed snapshot of the problem and the rationale of the study. Thank you very much.

Observation in the earlier review not addressed.

Comment9: Page4: Authors have used data from the National Family Health Survey (NFHS)-2015-16. Stunting becomes evident during 1st 1000 days of life. How the authors claim that the effect on the children born in 2015/2016 are sufficiently understood to be inferred?

Reply: The NFHS survey is a cross-sectional survey and the age of the children and the anthropometric measures are measures on the date of interview. Thus this study is limited to measure the effect of first 1000 days of life. At the same time this study is limited to measure the cohort effect except the age fixed effect. To mention, NFHS surveys of different rounds provide the estimates of undernutrition among under five children and the datasets are grossly used the socio-economic and demographic patterns and determinants of different child health parameters including child undernutrition within a cross-sectional framework. Similarly, this study also used the data information from NFHS-4 and examined the inequality in child nutrition using a novel decomposition approach.

Response against this observation is required to be captured appropriately in the “data” of Method section.

Reply: Like the DHS surveys, NFHS also only provides the wealth based measure of household’s economic wellbeing. This measure is based upon the asset information from the households. Thus it could be said that wealth measure of economic wellbeing captures the material affluence of the household.

What prevents the authors not to capture in the texts?

Reply: Thank you so much. We have changed the heading.

Please use the right English – “percentage”.

Authors are requested to capture the excerpts of revised texts while responding to the review comments in addition to specifying the respective line number with the page highlighting changes.

Reviewer #2: I am satisfied with the adjustments made to my previous comments. The discussion has been completely overhauled reflecting more scientific arguments.

7. PLOS authors have the option to publish the peer review history of their article (what does this mean?). If published, this will include your full peer review and any attached files.

Reviewer #1: No

Reviewer #2: No

---

## [Author Response · Author response to Decision Letter 1]

10 Oct 2024

The last revision was done. But this is a new submission with additional analysis and the data, results, discussion and references have been revised. And we look forward to get this manuscript published with PLOS ONE.

---

## [Editor Report · Decision Letter 2]

29 Oct 2024

Poverty induced inequality in nutrition among children born during 2010-21 in India

PONE-D-21-08355R2

Dear Dr. Khan,

We’re pleased to inform you that your manuscript has been judged scientifically suitable for publication and will be formally accepted for publication once it meets all outstanding technical requirements.

Kind regards,

Kannan Navaneetham, PhD

Academic Editor

PLOS ONE
---

## [Editor Report · Acceptance letter]

5 Nov 2024

PONE-D-21-08355R2 

PLOS ONE

Dear Dr. Khan, 

I'm pleased to inform you that your manuscript has been deemed suitable for publication in PLOS ONE. Congratulations! Your manuscript is now being handed over to our production team.

Kind regards, 

on behalf of

Prof. Kannan Navaneetham 

%CORR_ED_EDITOR_ROLE%

PLOS ONE